# ExAct: Teaching AI Agents to Explore with Reflective-MCTS and Exploratory Learning

**Xiao Yu**[1], **Baolin Peng**[2†], **Vineeth Vajipey**[1], **Hao Cheng**[2], **Michel Galley**[2]
**Jianfeng Gao**[2*] **& Zhou Yu**[1*]
[1]Columbia University, NY   [2]Microsoft Research, Redmond
{xy2437, zy2461}@columbia.edu
{baolinpeng, jfgao}@microsoft.com

## Abstract

Autonomous agents have demonstrated significant potential in automating complex multistep decision-making tasks. However, even state-of-the-art vision-language models (VLMs), such as GPT-4o, still fall short of human-level performance, particularly in intricate web environments and long-horizon tasks. To address these limitations, we present ExAct, an approach to combine test-time search and self-learning to build o1-like models for agentic applications. We first introduce Reflective Monte Carlo Tree Search (R-MCTS), a novel test-time algorithm designed to enhance AI agents' ability to explore decision space on the fly. R-MCTS extends traditional MCTS by 1) incorporating contrastive reflection, allowing agents to learn from past interactions and dynamically improve their search efficiency; and 2) using multi-agent debate to provide reliable state evaluation. Next, we introduce Exploratory Learning, a novel learning strategy to teach agents to search at inference time without relying on any external search algorithms. On the challenging VisualWebArena benchmark, our GPT-4o-based R-MCTS agent achieves a 6% to 30% relative improvement across various tasks compared to the previous state-of-the-art. Additionally, we show that the knowledge and experience gained from test-time search can be effectively transferred back to GPT-4o via fine-tuning. After Exploratory Learning, GPT-4o 1) demonstrates the ability to explore the environment, evaluate a state, and backtrack to viable ones when it detects that the current state cannot lead to success, and 2) matches 87% of R-MCTS's performance while using significantly less compute. Notably, our work demonstrates the compute scaling properties in both training - data collection with R-MCTS - and testing time. These results suggest a promising research direction to enhance VLMs' reasoning and planning capabilities for agentic applications via test-time search and self-learning.[1]

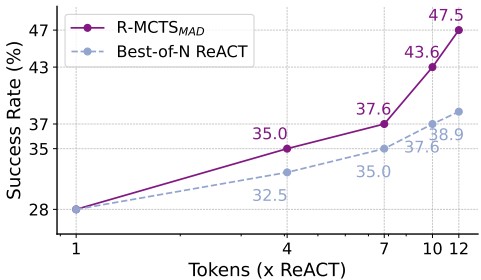 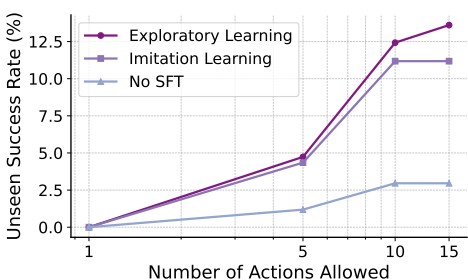

Figure 1: Our work yields compute scaling of GPT-4o with R-MCTS (left) and fine-tuned GPT-4o (right), for both training and testing, respectively. Left is evaluated on all 234 tasks from Classifieds in VisualWebArena, and right is evaluated on 169 unseen tasks from Classifieds.

---

[*]Equal Advisory Contribution; [†] Project Lead
[1]Code and data can be found at https://aka.ms/ExACT.

# 1 INTRODUCTION

Visual-Language Models (VLMs) have seen significant advancements, becoming increasingly powerful at processing and generating multimodal content. High-capacity models such as GPT-4o (OpenAI, 2024a), Gemini (Gemini, 2024), Phi-3 (Abdin et al., 2024), and Opus (Anthropic, 2024) have demonstrated promising understanding and reasoning abilities, paving the way for building VLM-based agents that can automate computer tasks such as software engineering (Yang et al., 2024; Wang et al., 2024), web navigation (Zhou et al., 2024b; Koh et al., 2024a) and mobile device control (Rawles et al., 2023; 2024). As many computer tasks require agents to interact with complex environments for long-term objectives, one critical challenge is to reason over extended periods and dynamically explore a large decision space. Current models often struggle at accessing a wide range of potential actions in these environments, and balancing exploration and exploitation in long-horizon tasks. Even GPT-4o-based agents significantly underperform humans, achieving success rates less than 20% compared to human performance of 89% (Koh et al., 2024a).

As demonstrated by OpenAI's o1 (OpenAI, 2024b) and Snell et al. (2024), increasing the number of thinking tokens as chain of thoughts at test time improves accuracy on tasks that require long-horizon planning and reasoning, such as math and coding. This naturally raises the question: Can we scale test-time computation to improve VLMs' decision-making capabilities in multistep planning and reasoning for *agentic* tasks? An intuitive strategy for agentic tasks, such as web navigation, is search. By exploring different actions and evaluating their consequences, models can gain a better understanding of the environment, reason about correct actions, and plan more effectively for long-term objectives. Recent work in this direction includes Tree of Thought (Yao et al., 2023a) and Search Agent (Koh et al., 2024b), which rely on best-first search in the form of BFS/DFS and A* search, respectively, to augment the VLM's decision-making process. However, these methods suffer from 1) a lack of balance between exploration and exploitation, which is crucial for handling complex tasks, and 2) an inability to learn from past interactions to improve future searches, which humans naturally do. More importantly, these works focus primarily on prompt-based methods and do not address how to *transfer* knowledge from search back into the model. This results in agents with high inference costs for complex tasks, as it requires the use of VLMs with search algorithms.

In this work, we explore how to effectively scale test-time compute to improve VLM agents, and efficiently transfer knowledge and experience acquired from search back to the model to enhance its reasoning and planning abilities. We present EXACT, an approach to combine test-time search and self-learning to build o1-like models for agentic applications. First, we introduce our Reflective Monte Carlo Tree Search (R-MCTS) agent. Our method extends the classic MCTS with two innovations: *contrastive reflection* to improve search quality in real time using high-quality reflections obtained from past experiences, inspired by conceptual models of human learning (Marton, 2014); and a *multi-agent debate* technique for reliable state evaluation, inspired by collaborative and iterative human evaluation approaches (van der Lee et al., 2019). Second, we propose Exploratory Learning, a method to teach agents to search at inference time *without* relying on any external search algorithms. In contrast to traditional Imitation Learning which trains the model with the optimal, final actions returned from the search process, Exploratory Learning teaches the models to explore the environment, evaluate a state, and backtrack to viable ones using search tree *traversals*.

Experiments on the challenging VisualWebArena (Koh et al., 2024a) show that our GPT-4o-based R-MCTS agent achieves a new state-of-the-art performance, with a 6% to 30% relative improvement across various environments compared to the previous state-of-the-art. Additionally, we find GPT-4o trained with Exploratory Learning demonstrates the ability to 1) explore, evaluate, and backtrack without any search augmentation; and 2) recover 87% of the search performance while reducing the inference cost by 2.7x. This represents an important step towards improving VLM's planning and reasoning capabilities for agentic applications by leveraging test-time search and self-learning.

# 2 BACKGROUND

## 2.1 NAVIGATING THE WEB

Decision making in a complex environment is typically formulated as a Partially Observable Markov Decision Process (POMDP) of $(\mathcal{S}, \mathcal{A}, \Omega, \mathcal{T})$, which consists of sets of states, actions, observations, and a transition function. In the context of web navigation, a *state* $s \in \mathcal{S}$ represents the entire environment's state including the current webpage and database states; an *action* $a \in \mathcal{A}$ is an ac-

tion the agent can express in natural language and execute in the environment (see Table 1); an *observation* is a textual or visual representation of the current webpage; and the *transition function* $\mathcal{T}(s, a) \to (s', o')$ executes an action $a$ in the environment and returns the next state and observation. For more details, we refer the reader to Zhou et al. (2024b); Koh et al. (2024a).

## 2.2 MONTE CARLO TREE SEARCH

In long-horizon tasks with a large action space, Monte Carlo Tree Search (MCTS) (Silver et al., 2017; Świechowski et al., 2022) is a popular method to improve decision making. MCTS explores multiple actions, evaluates their consequences, and returns the best action for execution after conducting a large number of simulations. We briefly describe the MCTS algorithm here and refer to Appendix A.1 for more details. Given a state to search from, MCTS iteratively constructs a tree by: 1) *selecting* an action $a$ to explore/exploit using Upper Confidence Tree Bound (UCT) (Kocsis & Szepesvári, 2006; Rosin, 2011); 2) *expanding and evaluating* the selected action by simulating the next state $\mathcal{T}(s, a) \to (s', o')$ and evaluating current progress; 3) *backpropagating* the evaluation to update the tree's value estimates of quality of the executed action. This search process is repeated until the search budget is exhausted, and the most-visited (Silver et al., 2017) or the highest-value action (Kocsis & Szepesvári, 2006) is returned.

| Action Type $a$ | Description |
| --- | --- |
| click [elem] | Click on elem. |
| hover [elem] | Hover over elem. |
| type [elem][text] | Type text into elem. |
| press [key_comb] | Press a key combo. |
| new_tab | Open a new tab. |
| tab_focus [index] | Focus on the i-th tab. |
| tab_close | Close current tab. |
| goto [url] | Go to url. |
| go_back | Click back. |
| go_forward | Click forward. |
| scroll [up/down] | Scroll up or down. |
| stop [answer] | End with an output. |

Table 1: Action space for web navigation defined in VisualWebArena (Koh et al., 2024a).

## 2.3 SETUP

We now present the setup commonly used by VLMs for agentic tasks, e.g., that of VisualWebArena (Koh et al., 2024a). A web agent begins by receiving a task $g$ in natural language (and images) and starting webpage $o_0$. Then, at each time step $t$: 1) the agent generates an action $a_t \in \mathcal{A}$ based on the last state $s_t$, which encodes information from past actions and observations; and 2) $a_t$ is executed, transiting to a new state $\mathcal{T}(s_t, a_t) \to (s_{t+1}, o_{t+1})$, and observation $o_{t+1}$ is returned to the agent. This agent-environment interaction continues until either the maximum number of steps $T$ is reached, or the agent issues a STOP action to terminate the episode. Finally, a terminal reward $r_T \in \{0, 1\}$ is returned based on the final answer or the last environment state $s_T$.

In practice, visual agents are often implemented using a large vision-language model (VLM) such as GPT-4o (OpenAI, 2024a). Common approaches include directly prompting the VLM as a policy function to generate an action $\pi(\cdot) \to a$ given a task and previous actions and observations (Yao et al., 2023b; Koh et al., 2024a); or to additionally construct value functions $V(\cdot)$ and conduct simulation-based searches before returning an action (Yao et al., 2023a; Zhou et al., 2024a; Koh et al., 2024b). We detail these methods below.

**VLM as a Policy Function** Given a task $g$ and an trajectory $\tau = \{o_0, a_0, \dots, \dots, a_{t-1}, o_t\}$, a policy $\pi(g, \tau) \to a_t$ returns the next action $a_t$ to execute. Popular methods such as ReACT (Yao et al., 2023b) use a VLM as policy by directly prompting it for an action, after optionally generating a reasoning step. Then, the environment executes action $a_t$ and returns a new observation $o_{t+1}$. This process repeats until a STOP action is issued or the maximum number of steps is reached.

**VLM as a Value Function** To augment an agent's decision-making process, many recent works implement a value function $V(\cdot)$ to guide the VLM to predict better actions. Given a trajectory $\tau$, a value function estimates the expected success rate given the current state $s_t$. However, since state $s_t$ of the environment (e.g., including database information) may not always be accessible to the agent, value functions are often implemented using the current *observed* trajectory $\tau$ and the task description $g$ instead: $V(g, \tau) \to [0, 1]$. In this work, we consider two value functions. A *single-agent* value function (Yao et al., 2023a; Koh et al., 2024b) directly prompts a VLM with all inputs

(task $g$, trajectory $\tau$) to generate a value estimate $v_{\text{SA}}$:

$$v_{\text{SA}} = \text{VLM}\,(g, \tau) \in [0, 1],$$

and a value function based on multi-agent debate, which we describe in the next section.

## 3 METHOD

We propose EXACT, an approach to teach AI agents to explore via test-time search (Section 3.1) and self-learning (Section 3.2).

### 3.1 REFLECTIVE MCTS

Web tasks are highly diverse as different websites have different layouts and functionalities (e.g., Amazon versus Reddit), and almost every webpage is unique. Although search-augmented agents have shown promising results in web navigation tasks, their performance is limited by the complexity of web environments and their inability to quickly adapt to unseen environments.

We propose Reflective Monte Carlo Tree Search (R-MCTS), an extension of classic MCTS that improves the agent's decision making process on the fly by incorporating reflection over its past task executions, and state estimations using multi-agent-debate. We present a high-level overview of an R-MCTS agent in Figure 2, and a pseudo-code description in Appendix A.2. Similarly to classic MCTS, an R-MCTS agent during **inference** predicts an action $a_t$ by iteratively building a search tree that explores the current decision space (i.e., looping over selection, expansion, simulation, and backpropagation steps). Unlike classic MCTS, R-MCTS also iteratively **improves** the search process by 1) using contrastive reflection to identifying past mistakes/successes; and 2) updating the agent's policy and value functions (in-context) to improve its future task executions during infer-

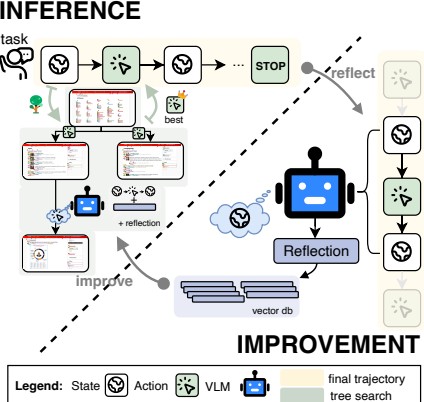

Figure 2: Overview of an R-MCTS Agent. We omit value function reflection for brevity.

ence. This improvement step helps R-MCTS to avoid repeating the same mistakes and to explore more promising actions when faced with similar situations in the future. For clarity, we describe this improvement process in the context of *policy* reflection, since value reflection is highly similar[2].

**Contrastive Reflection**   Policy (or value) function consists of three steps: 1) error attribution; 2) constrastive learning; and 3) memorization of reflection. Given a (long-horizon) trajectory $\tau$, this process intuitively corresponds to first identifying the most 'erroneous' action, then analyzing the error by comparing what the agent *expects* to achieve with what *actually happens*, and finally storing this knowledge by remembering the task and state where the error occurs. This process is inspired by human cognitive learning (Marton, 2014), where contrasting our mental model of the world with the reality is an effective way to obtain new knowledge.

Formally, given a task $g$ and trajectory $\tau$, we first identify $n_\pi$ most erroneous actions (and $n_V$ value estimates for value reflection) for reflection, based on the difference between the VLM's *predicted* future success $V$ and the search-tree's *simulated* future success $Q$:

$$\text{error}_\pi(a_t|\tau) = |V(o_{t+1}) - Q(o_t, a_t)|, \quad \text{error}_V(o_t|\tau) = |V(o_t) - Q(o_{t-1}, a_{t-1})|.$$

Note that this form of comparing value function $V$ and action-value function $Q$ is similar to Temporal Difference Error used in reinforcement learning (Sutton & Barto, 2018). For each of the erroneous actions $\{\tilde{a}_1, ..., \tilde{a}_{n_\pi}\}$, we then prompt the VLM to identify reasoning mistakes and gain understanding of navigating in the specific environment. We achieve this by contrastive learning[3]:

---

[2]For value reflection, we reflect on states for erroneous state estimations instead of actions. Then, we prompt the VLM to predict the expected future actions, and generate a reflection contrasting the agent's actual actions.

[3]For simplicity, we use the accessibility tree representation of $o$ in text modality throughout this process.

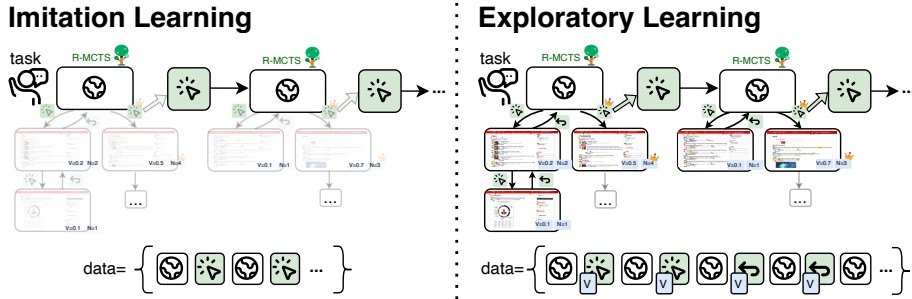

Figure 3: Given a trajectory from R-MCTS, *Imitation Learning* removes intermediate search trees and directly trains GPT-4o to learn from the final executed actions; *Exploratory Learning* flattens tree traversals into a single trajectory and trains GPT-4o to explore, evaluate, and backtrack.

we first 1) prompt the VLM to predict the expected transition $\hat{o}_{t+1}$ after executing $\tilde{a}_t$, and 2) prompt the VLM to generate a reflection by contrasting the current $o_t$, $\tilde{a}_t$, the expected $\hat{o}_{t+1}$ and real $o_{t+1}$:

$$\hat{o}_{t+1} = \text{VLM}_{\text{simulate}}(g, \{o_0, a_0, ..., o_t, \tilde{a}_t\}), \quad \text{reflect}(\tilde{a}_t | g, \tau) = \text{VLM}_{\text{reflect}}(g, o_t, \tilde{a}_t, \{o_{t+1}, \hat{o}_{t+1}\}).$$

Finally, to help the agent memorize this reflection at test-time, we embed the reflection using the current task and state $(g, o_t)$ and store it in a vector database.

**Improvement** Then, given a new task and a new observation $(g^{\text{new}}, o_t^{\text{new}})$ to perform inference, we improve the policy (and value) function by 1) retrieving the most relevant reflections using cosine similarity of their embeddings[4], and 2) appending them into the agent's current context. For clarity, we also illustrate this reflection and improvement process in Figure 5 (Appendix A.2).

**Multi-Agent Value Function** In addition to the reflection-improvement loop, we also experiment with using multi-agent value function to provide more reliable state estimates. Intuitively, this method offers 1) a more holistic view of the current state to mitigate mistakes caused by oversights; and 2) stronger reasoning elicited by collaborative/adversarial incentives (Bowman et al., 2022).

Formally, a *multi-agent* value function prompts multiple VLMs to each generate a value estimate $v^i$, and then aggregate all estimates $\{v^1, v^2, ...\}$ to produce a final estimate $v_{\text{MA}}$:

$$v^i = \text{VLM}_i(g, \tau) \in [0, 1] \quad \text{and} \quad v_{\text{MA}} = \text{aggregate}(g, \tau, \{v^1, v^2, ...\}) \in [0, 1].$$

In this study, we implement the multi-agent value function using *multi-agent-debate* (MAD). Given a task $g$ and a trajectory $\tau$, MAD prompts two VLMs to generate two opposing arguments for the current value estimate (i.e., why the current state is promising/not promising), and then aggregates the two arguments using another VLM to obtain a final judgement:

$$\text{aggregate}(g, \tau, \{v^1, v^2, ...\}) = \text{VLM}_{\text{judge}}(g, \tau, \{v^1, v^2, ...\}).$$

For the prompts used by contrastive reflection, improvement, and MAD, please refer to Appendix E.2.

### 3.2 EXPLORATORY LEARNING

Search-augmented agents such as R-MCTS improve performance at the expense of increased test-time computation. Using GPT-4o powered R-MCTS as an example, we explore how to teach agents to search and explore at inference time without relying on any external search algorithms. We propose **Exploratory Learning** (EL), a learning strategy to teach GPT-4o to explore, evaluate, and backtrackon using all tree traversals that happened during search. This is contrast to the traditional **Imitation Learning** (IL) approach, which directly fine-tunes GPT-4o on the actions returned by the search algorithm. We illustrate these two methods in Figure 3.

---

[4]For simplicity, we directly concatenate $(g^{\text{new}}, o_t^{\text{new}})$ into a single string and feed into an embedding model (text-ada-003-small, OpenAI (2024c)).

Given a task $g$, a trajectory $\tau = \{o_0, a_0, ..., a_T\}$, and all search trees $\text{Tree}(o_0), .., \text{Tree}(o_T)$ obtained during R-MCTS, Imitation Learning trains GPT-4o to learn an improved policy by directly fine-tuning on the actions executed in $\tau$ (i.e., best actions selected by search). This method is simple, but relies on the model itself to learn the implicit reasoning process. Exploratory Learning improves GPT-4o's decision making ability by using both $\tau$ and tree traversals $\text{Tree}(o_0), .., \text{Tree}(o_T)$. Specifically, given a tree $\text{Tree}(o_i)$, we 1) replay the tree search process[5] to obtain observations $o$, actions explored or backtracked $a$, and the estimated values $v$; 2) combine value estimation and action into a single action $a \leftarrow (v, a)$, and 3) append them to form a single trajectory $\tau_i'$. We then repeat this for all trees, and obtain a single trajectory $\tau' = \{\tau_0', ..., \tau_T'\}$ used to train GPT-4o.

## 4 EXPERIMENTS

We primarily experiment on VisualWebArena (VWA), a benchmark designed to evaluate multimodal agents performance on across a wide range of web navigation tasks and environments. To further measure the generalizability of our method, we also evaluate on other popular benchmarks such as OSWorld and relevant domains from WebArena. We describe the experimental setup, baselines, and other relevant implementation details below.

**Benchmarks** VisualWebArena (Koh et al., 2024a) is a large-scale benchmark consisting of 910 tasks across three different web environments: Classifieds, Reddit, and Shopping. All tasks in VWA require visual understanding of webpage content to solve effectively, and 25.2% of the tasks also include images as input (e.g., help me make a post selling this item + an image of a phone). We host all web environments locally and evaluate on all 910 tasks, unless otherwise specified. We follow (Koh et al., 2024a;b) and use Set-of-Mark (Yang et al., 2023b) augmented web screenshots as the agent's input. An example is presented in Appendix E.1.

Beyond VWA, we also consider OSWorld (Xie et al., 2024), an diverse benchmark consisting of 369 open-ended computer tasks that involves real web and desktop apps in open domains, OS file I/O, and workflows spanning multiple applications; and relevant domains (i.e., GitHub) from WebArena (Zhou et al., 2024b) to further enhance our evaluation.

**Baselines** We mainly compare R-MCTS against direct prompting methods as well as search-augmented agents backed by various algorithms. For direct prompting methods, we consider REACT (Yao et al., 2023b), which prompts a VLM to generate the reasoning step before generating the action. For search, we consider Tree of Thought (TOT, Yao et al. (2023a), which uses BFS or DFS to explore the decision space and returns the best action found according to $V$; SEARCH AGENT (Koh et al., 2024b), which is a best-first method inspired by A* search; and MCTS (Silver et al., 2017; Zhou et al., 2024a; Yu et al., 2023), which uses MCTS to determine the best action to take.

For evaluations on OSWorld, please refer to appendix C for more details.

**Policy and Value Implementation** Search methods require a policy that can generate up to $b$ actions for exploration, and a value function that returns a value between 0 and 1. We follow Koh et al. (2024b) to implement such a policy by 1) sampling up to 20 responses from the VLM with a temperature of 1.0 and a top-$p$ of 0.95, using nucleus sampling (Holtzman et al., 2020); 2) aggregating the counts of each unique action; and 3) returning the top-$b$ actions with the highest counts. We implement the single-agent value function $V_{\text{SA}}$ by 1) prompting the VLM with a multiple-choice based prompt; 2) sampling 20 responses with temperature of 1.0 and top-$p$ of 0.95; 3) converting the selected choices into a numeric value; and 4) returning the average value of all responses. We implement the multi-agent value function $V_{\text{MA}}$ by 1) prompting the VLM twice to generate reasons why the current state is promising/unpromising; 2) prompt the VLM again to generate a final judgement with a multiple-choice-based prompt; and 3) converting the selected choice into a numeric value. We use GPT-4o (2024-05-13) (OpenAI, 2024a) as the VLM for all methods as it has been widely used in the development of state-of-the-art agents (Koh et al., 2024b; Wang et al., 2024)

**Search Parameters** On VisualWebArena, we follow Koh et al. (2024b) to compare all search methods with a breadth $b$ and depth $d$ limit of 5 per tree, and to stop task execution after a maximum

---

[5]This replay stops at the first time when the search process reached the best action.

of 5 actions. Since different algorithms behave differently *during* search, we also set a maximum search time of 5 minutes per state. For MCTS-based algorithms, we use a UCT bound with $c_p = 1.0$ for action selection, and use the most-visited child for selecting the best action (Silver et al., 2017). On OSWorld, we follow other official results and stop task execution after a maximum 15 steps. Since this is significantly longer than the evaluation on VWA, we set a max node limit of 60, and set $b = 10$ to be decaying exponentially per depth when running R-MCTS.

## 4.1 R-MCTS RESULTS

| Search | Value | Classifieds (VWA) | | Reddit (VWA) | | Shopping (VWA) | | GitLab (WA) | |
|---|---|---|---|---|---|---|---|---|---|
| | | Tokens | Success | Tokens | Success | Tokens | Success | Tokens | Success |
| ✗ (REACT) | - | 1x | 28.6% | 1x | 13.8% | 1x | 23.2% | 1x | 11.3% |
| TOT$_{\text{BFS}}$ | SA | 3.2x | 30.7% | 4.7x | 19.5% | 4.3x | 29.2% | 4.6x | 14.1% |
| TOT$_{\text{DFS}}$ | SA | 3.1x | 30.3% | 4.2x | 16.7% | 4.0x | 28.3% | 4.3x | 14.1% |
| SEARCH AGENT | SA | 4.2x | 33.8% | 5.4x | 21.9% | 5.1x | 30.3% | 3.0x | 13.8% |
| MCTS | SA | 7.2x | 37.6% | 9.5x | 23.8% | 8.9x | 29.4% | 5.7x | 19.9% |
| R-MCTS | SA | 7.3x | 40.2% | 7.3x | 25.2% | 7.6x | 31.9% | 6.0x | 20.9% |
| R-MCTS | MAD | 7.4x | **41.0**% | 9.7x | **28.7**% | 10.1x | **32.3**% | 8.8x | **23.5**% |

Table 2: Comparing different agent's token consumption and performance on VisualWebArena (VWA). For more domain diversity, we also evaluate GitLab tasks from WebArena (WA). We show average token used per task using REACT as a baseline.

We summarize the performance of R-MCTS and other search-augmented agents evaluated on VisualWebArena (VWA) in Table 2. We observe that all search-based methods deliver significant improvements across the VWA environments, albeit with increased test-time compute measured by the number of tokens. Among these methods, MCTS consistently outperforms its counterparts. This is likely due to MCTS's ability to balance exploration and exploitation through UCT, whereas other methods primarily rely on $V$ to guide the search. Additionally, our R-MCTS agent sets a new state-of-the-art performance benchmark across all VWA environments, with relative improvements ranging from 6% to 30% over the previous best-performing method, Search Agent. When using a single-agent value function, R-MCTS improves upon MCTS by an average of 2.2 points. Furthermore, with the integration of multi-agent debate, R-MCTS achieves an additional 1.6-point improvement on average. These findings highlight a promising approach to scaling test-time compute in agentic applications by integrating search, reflection, and multi-agent debate strategies. The comprehensive leaderboard for VisualWebArena can be found in Appendix Table 10.

We also extend our R-MCTS agent on OSWorld. Please refer to Appendix C for more details.

## 4.2 LEARNING RESULTS

Next, we evaluate EXACT: teaching GPT-4o to explore and improve its performance using knowledge and experience gained from R-MCTS. Since GPT-4o fine-tuning does not support images[6], we evaluate all methods with a text-only modality from VWA. We mainly consider training and evaluating on 234 tasks from the Classifieds environment, as GPT-4o finetuning is costly. For simplicity, we refer to R-MCTS with a multi-agent-debate value function $V_{\text{MAD}}$ as R-MCTS$_{\text{MAD}}$. Note that EXACT and REACT use similar prompts, with slight modifications in the system prompts to encourage exploration

**Training Data** We first run R-MCTS$_{\text{MAD}}$ on Classifieds, and then sampled 65 trajectories for Imitation Learning according to the estimated values of final success. However, since search tree traversals are typically long, we further remove trajectory that results in more than 20 actions, producing 35 trajectories. We format these data into multi-turn chats, and use OpenAI finetuning API for training.

**Quantitative Results** We present the comparison results with REACT in Table 3. Our findings indicate that both EXACT, trained with Imitation Learning and Exploratory Learning, outperform

---

[6]At the time of the study, GPT-4o vision-finetuning was not yet supported.

| Method | Model | Search | Max Steps | Training | Tokens | Success | | |
|--------|-------|--------|-----------|----------|--------|---------|---------|---------|
| | | | | | | Seen | Unseen | Overall |
| REACT | GPT-4o-mini | ✗ | 20 | ✗ | 1.0x | - | - | 20.9% |
| REACT | o1-mini | ✗ | 20 | ✗ | 1.0x† | - | - | 23.9% |
| REACT | GPT-4o | ✗ | 5 | ✗ | 1.0x | - | - | 22.2% |
| REACT | GPT-4o | ✗ | 20 | ✗ | 2.4x | - | - | 22.2% |
| EXACT | GPT-4o | R-MCTS$_{MAD}$ | 20 | ✗ | 9.6x | - | - | 32.1% |
| EXACT | GPT-4o w/ IL | ✗ | 20 | 65 | 2.5x | 80.0% | 12.4% | 31.2% |
| EXACT | GPT-4o w/ EL | ✗ | 20 | 35 | 3.6x | 80.0% | 18.6% | 27.8% |

Table 3: Comparing different model's performance evaluated in the Classifieds environment. All non-search methods are based on direct prompting. †While total token consumption is similar, o1-mini used 20x more output tokens (reasoning + output) than GPT-4o-mini's output tokens (output).

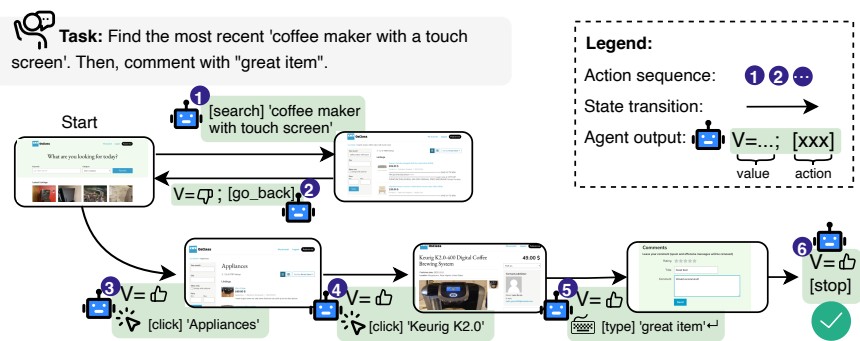

Figure 4: After Exploratory Learning, GPT-4o demonstrates the ability to explore, evaluate, and backtrack without augmenting with search algorithms.

REACT (based on the untrained GPT-4o) by a considerable margin, even without utilizing any search algorithm. Compared to R-MCTS$_{MAD}$, both methods achieved over 85% of the performance while reducing token usage by up to 4x. In addition, EXACT with EL achieves a significantly higher success rate on unseen tasks, demonstrating the effectiveness of exploratory learning. These results suggest that even powerful models like GPT-4o can benefit greatly from training with agentic data.

**Qualitative Results** With Exploratory Learning, we find that GPT-4o is able to perform better on unseen tasks by performing exploration, state evaluation, and backtracking without relying on any search algorithm. We illustrate an example in Figure 4: after attempting to directly search for "coffee maker with a touch screen", the Exploratory Learning trained-GPT-4o finds that the search function in Classifieds may be unreliable (top search result is a sofa) and backtracked to use category filters instead.

### 4.3 COMPUTE SCALING RESULTS

We explore whether R-MCTS$_{MAD}$ *and* the fine-tuned model after the self-learning stage has the property of test-time computing scaling, i.e., whether performance can increase when more test-time computes, such search tokens, are allocated. To study this, we follow the same experiment setup in our main results in Section 4.1, but vary the search budget to allow R-MCTS$_{MAD}$ for $\{2, 5, 10, 15\}$ nodes per search tree. We then compare performance (*Success Rate*) and token consumption per task compared to REACT (*Tokens*) in Figure 1a. Since searching with more nodes significantly increases time and cost, we report results by testing on all 234 tasks in the Classifieds environment.

As shown in Figure 1, our proposed R-MCTS and Exploratory Learning exhibit strong scaling properties for both train-time and test-time compute. Particularly, in Figure 1 left, we find that R-MCTS$_{MAD}$ significantly improves performance when allocated with more test-time compute: using 15 nodes per tree, R-MCTS$_{MAD}$ achieves a relative improvement of 66% compared to REACT. Furthermore, Figure 1 right also illustrates that *without* search, both Imitation Learning and Exploratory

Learning substantially boost the performance of an untrained GPT-4o when allowed more actions per task. Notably, Exploratory Learning demonstrates a more favorable scaling trend in test-time compute compared to Imitation Learning, as it is trained to learn to search. These results highlight the potential for self-improvement via reflective search and self-learning in agentic applications.

## 5 ANALYSIS

### 5.1 SUCCESS RATE BREAKDOWN

In addition to having tasks in different environments, VWA labels these task for *action difficulty*. These are labeled by estimating the number of steps an average human would take to complete the task: easy (1-3 steps), medium (4-9 steps), and hard (10 steps or more). We evaluate all methods using the same setup as in our main results in Section 4.1. We present the results in Table 4.

| Search | Easy | Medium | Hard |
|---|---|---|---|
| ✗ (REACT) | 42.4% | 20.9% | 11.6% |
| ToT$_{BFS}$ | 46.4% | 28.0% | 14.7% |
| ToT$_{DFS}$ | 46.8% | 26.5% | 12.8% |
| Search Agent | 45.9% | 29.6% | 18.4% |
| MCTS | 47.8% | 32.5% | 16.5% |
| R-MCTS | **50.7%** | 33.6% | 19.9% |
| R-MCTS$_{MAD}$ | 49.3% | **34.1%** | **23.6%** |

Table 4: VWA performance by difficulty.

**R-MCTS improves performance across difficulties**    Compared to prior best, we find R-MCTS agent improved performance by an avearge of 2.5% absolute across all difficulties. This shows that the search and improvement loop in the R-MCTS agent not only benefits planning in long-horizon tasks, but also enhances robustness in solving easier tasks.

**Multi-Agent-Debate benefits difficult tasks**    When using a multi-agent value function (R-MCTS$_{MAD}$), performance is further improved in medium and hard tasks. We believe this is because multi-agent debate encourages a more critical view of the current state, and making it less error-prone in long horizon tasks. However, in easier tasks, we find such method can suffer from "over-thinking" by trying to avoid non-existent issues, which can slightly reduce performance.

### 5.2 ABLATION STUDIES

We perform ablation studies for each of component in R-MCTS$_{MAD}$ to understand their contributions to the overall performance. In Table 5, we report the overall performance across all 910 tasks in VWA when removing 1) reflection from value function; 2) reflection from policy; and 3) the search algorithm. We find that reflections for policy improvement are more crucial than reflections for value function, and that search is essential for competitive performance.

| Method | Overall | |
|---|---|---|
| | Token | Success |
| R-MCTS$_{MAD}$ | 9.3x | 33.7% |
| - *Value refl.* | 9.0x | 32.9% |
| - *Policy refl.* | 8.6x | 30.2% |
| - *Search* | 1.0x | 21.9% |

Table 5: Ablation studies on VWA.

### 5.3 QUALITATIVE ANALYSIS

To better understand the search process in R-MCTS$_{MAD}$, we manually inspect 80 randomly sampled trajectories and their corresponding search trees. We compare against trajectories generated by SEARCH AGENT and analyze errors made by our method.

**Exploration and Reflection improves action quality**    Amongst the 80 samples, we find that R-MCTS$_{MAD}$ outperforms SEARCH AGENT in 12.5% of the tasks. We find that this is primarily due to R-MCTS$_{MAD}$'s ability to improve its policy (60%) using the reflection-improvement process (Section 3.1); and its ability to balance exploration-exploitation (30%) using UCT instead of best-first.

**GPT-4o still lacks finegrained image and web understanding**    In Figure 6, we categorize all errors made by R-MCTS$_{MAD}$ during its search process by 1. failing to correctly format the final answer (*answer formatting error*); 2. errors caused by the agent-environment interaction (*environmental error*); 3. errors caused by GPT-4o's inability to perform fine-grained image understanding (*model image understanding*); and 4. errors caused by GPT-4o's limited web understanding/reasoning ability (*model web understanding*). For more details, please refer to Appendix D.1. Our analysis reveals

that most of the errors are caused by the backbone VLM's inability to understand the provided web-page screenshot/product images. Examples include failing to correctly identify the item on *a-th row and b-th column* on a shopping page; and misunderstanding that the cheapest product shown on the *current page* (sorted by most recently listed) is the cheapest product available *on the entire site*. We believe these model issues are fundamental to many current agents that rely on prompting and suggest that model training (alike Section 3.2) is essential to further improve agent's performance.

## 6 RELATED WORK

**Language-Guided Agents** Advances in large language models (LLMs) have motivated many recent works to re-purpose LLMs for automating agentic tasks (Nakano et al., 2022; Kim et al., 2023; Xi et al., 2023). These methods include prompting LLM directly as a policy (Hong et al., 2023; Yao et al., 2023b; Sridhar et al., 2023; Yang et al., 2023a), as a component of a manually designed agentic framework (Park et al., 2023; Sumers et al., 2024), combined with manually crafted heuristics/prompt designs to improve performance (Fu et al., 2024; Sodhi et al., 2024), or combined with tool uses for question answering/code generation (Yang et al., 2024; Xia et al., 2024). For tasks that requires visual understanding, many recent work have thus swapped the prompting LLMs to prompting visual language models (VLMs) such as GPT-4o (Zheng et al., 2024; Koh et al., 2024a). However, these methods typically suffer in long-horizon tasks that requires planning and error recovery. Our work thus augments VLMs agents with search algorithms such as MCTS (Silver et al., 2017) to complete tasks in complex and realistic web environments.

**Augmenting Agent with Search/Reflection** Using LLMs or VLMs to solve long-horizon tasks such as web navigation is a highly challenging task (Liu et al., 2023; Zhou et al., 2024b; Koh et al., 2024a). Many recent work thus also explored various strategies to improve agent's decision making process, such as: iteratively prompting the model to improve its own output (Madaan et al., 2023; Yu et al., 2024; Shinn et al., 2023); and augmenting agent's decision process using search algorithms such as BFS/DFS (Yao et al., 2023a), best-first search (Koh et al., 2024b), and MCTS (Yu et al., 2023; Zhou et al., 2024a). However, many of these methods primarily focus on text-based environments such as question answering and programming, and do not involve substantial interaction with a complex environment. In contrast, our work focuses on 1) conducting MCTS in long-horizon, agentic tasks in a realistic web environment; 2) incorporating multi-agent-debate value function to improve state evaluation; and 3) using contrastive reflection to learn from past successes and failures in long execution trajectories.

**Training Agents** Besides improving agent's performance at test-time, many works also explored methods to train the backbone LLM/VLM. Examples include Mind2Web (Deng et al., 2023), Fire-ACT Chen et al. (2023), AgentInstruct (Zeng et al., 2023), WebGum (Furuta et al., 2024), AutoWebGLM (Lai et al., 2024), and more (Zhang et al., 2024b; Liu et al., 2024; Zhang et al., 2024a). These methods rely on collecting human or machine generated trajectories based on direct prompting, and perform supervised training to improve model's performance. To our knowledge, we are the first to train GPT-4o using trajectories produced by MCTS, and to teach GPT-4o to explore, evaluate, and backtrack in complex web environments through training on MCTS tree traversal.

## 7 CONCLUSION

In this work, we first introduced R-MCTS agent, a search-augmented agent powered by 1) MCTS to explore, evaluate, and backtrack in complex and realistic web environments; 2) a multi-agent-debate value function for reliable state evaluation; and 3) contrastive reflection to learn from past successes and failures in long execution trajectories. Then, we proposed methods to transfer knowledge acquired from search back to GPT-4o by 1) training on best actions returned by tree search (Imitation Learning); and 2) training on the entire MCTS tree traversal (Exploratory Learning). Experiments on the challenging VisualWebArena benchmark show that R-MCTS achieves a new state of the art, and that our finetuned GPT-4o agent begins to display test-time compute scaling properties similar to search algorithms. We believe our work presents a promising direction to combine test-time search and model self-learning to develop the next-generation autonomous agents.

## 8 LIMITATIONS

**High cost of scaling test-time compute** Although MCTS-based agents achieve strong performance on benchmarks such as VWA, they consume nearly 10x more tokens compared to REACT. This presents a clear trade-off between performance and cost/time, and may limit the practicality of deploying such agents in real-world applications. However, these search-augmented agents can be used to *curate training data* for VLMs without human labels, which can in turn help develop stronger VLMs. Our work presents a promising step in this direction using Imitation Learning and Exploratory Learning, and we believe such a self-learning loop is crucial to build stronger agents.

**Long trajectories from tree traversal** In long-horizon tasks, R-MCTS agents require a large amount of simulation to identify optimal actions, resulting in long tree traversals. Training on these long traversals can be costly, and may be difficult for models to learn from. Nevertheless, we believe this issue can be iteratively mitigated by using self-learning to improve model capability, and by conducting search using the improved policy/value functions. We leave explorations for future work.

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

# A    ADDITIONAL IMPLEMENTATION DETAILS

## A.1    MONTE CARLO TREE SEARCH

We describe the Monte Carlo Tree Search (MCTS) algorithm used in our work in Algorithm 1. We use the same MCTS search implementation for all relevant methods (MCTS, R-MCTS, and R-MCTS$_{\text{MAD}}$ agent). For clarity, we abuse the notation $s$ to represent *observations* in this section, in order to comply with notations used in other works (Świechowski et al., 2022).

---

**Algorithm 1** MCTS

---

**Require:** VLM policy $\pi(g, \tau)$
**Require:** VLM value $V(g, \tau)$
**Require:** environment $\mathcal{T}$
**Require:** goal $g$ and current observation $s$
**Require:** actual trajectory so far $\tau$ (before $s$)
 1: **while** search budget is not exhausted **do**
 2:     $s_{\text{curr}} \leftarrow s$
 3:     $\tau_{\text{curr}} \leftarrow \tau \cup s$
 4:     *// selection*
 5:     **while** $s_{\text{curr}}$ is not a leaf node **do**
 6:         $a \leftarrow \arg\max_a Q(s_{\text{curr}}, a) + U(s_{\text{curr}}, a)$
 7:         $s_{\text{curr}} \leftarrow \mathcal{T}(s_{\text{curr}}, a)$
 8:         $\tau_{\text{curr}} \leftarrow \tau_{\text{curr}} \cup \{a, s_{\text{curr}}\}$
 9:     **end while**
10:     *// expansion*
11:     $\{a^1, a^2, ..., a^N\} \leftarrow \pi(g, \tau_{\text{curr}})$
12:     *// evaluation*
13:     $v \leftarrow V(g, \tau_{\text{curr}})$
14:     *// back-propagation*
15:     **while** $s_{\text{curr}} \neq s$ **do**
16:         $s_{\text{curr}} \leftarrow \text{parent}(s_{\text{curr}}, a_{\text{curr}})$
17:         update $Q, N$ using $v$ with Equation (2)
18:     **end while**
19: **end while**
20: *// prediction*
21: $a^* \leftarrow \arg\max_a N(s, a)$
22: **return** $a^*$

---

During selection, we follow Silver et al. (2017) and use a variant of PUCT (Rosin, 2011) to balance exploration and exploitation:

$$U(s, a) = c_p \cdot P(a|s) \cdot \frac{\sqrt{\sum_b N(s, b)}}{1 + N(s, a)} \tag{1}$$

where $c_p$ is a hyperparameter controlling the exploration rate, $P(a|s)$ is the VLM's prior probability of generating action $a$ in state $s$, and $N(s, a)$ is the state visit count. We use $c_p = 1.0$ in our experiments.

During evaluation, we prompt the VLM as a value function using the process described in Section 2.3. During back-propagation, we update the search tree's action-value estimate $Q$ and visitation count $N$ using running average:

$$Q(s, a) \leftarrow Q(s, a) + \frac{v - Q(s, a)}{N(s, a)}; \quad N(s, a) \leftarrow N(s, a) + 1. \tag{2}$$

---

**Algorithm 2** $R$-MCTS Agent

---

**Require:** VLM policy $\pi(g, \tau)$
**Require:** VLM value $V(g, \tau)$
**Require:** environment $\mathcal{T}$
**Require:** list of tasks to complete $G$
1: **for** task $g$ in $G$ **do**
2:     *// inference loop*
3:     $\tau \leftarrow \{s_0\}$
4:     tree $\leftarrow \{s_0\}$
5:     **while** task not terminated **do**
6:         **while** search budget not exhausted **do**
7:             *// selection*
8:             $s_t \leftarrow$ selection(tree)
9:             *// expansion*
10:            reflections$_\pi \leftarrow$ retrieve(db$_\pi$, $g$, $s_t$)
11:            $a_t^1, ..., a_t^k \leftarrow \pi(g, \tau, \text{reflections}_\pi)$
12:            *// simulation*
13:            $s_{t+1} \leftarrow \mathcal{T}(s_t, a_t)$
14:            reflections$_V \leftarrow$ retrieve(db$_V$, $g$, $s_{t+1}$)
15:            $v \leftarrow V(g, \{s_0, a_0, ..., s_{t+1}\}, \text{reflections}_V)$
16:            *// back-propagation*
17:            tree $\leftarrow$ update(tree, $s_{t+1}$, $v$)
18:         **end while**
19:         $a_t \leftarrow$ best_action(tree)
20:         $s_{t+1} \leftarrow \mathcal{T}(s_t, a_t)$
21:         $\tau \leftarrow \tau \cup \{a_t, s_{t+1}\}$
22:         tree $\leftarrow \{s_{t+1}\}$
23:     **end while**
24:     $r \leftarrow$ task success/failure
25:     *// improvement loop*
26:     reflections$_\pi$, reflections$_V \leftarrow$ reflect(VLM, $g$, $\tau$)
27:     db$_\pi \leftarrow$ update(reflections$_\pi$)
28:     db$_V \leftarrow$ update(reflections$_V$)
29: **end for**

---

| **Algorithm 3** Policy Reflection | **Algorithm 4** Value Reflection |
|---|---|
| **Require:** VLM used during search | **Require:** VLM used during search |
| **Require:** Terminated task $g$ and trajectory $\tau$ | **Require:** Terminated task $g$ and trajectory $\tau$ |
| **Require:** Search tree statistics $Q, V$ | **Require:** Search tree statistics $Q, V$ |
| **Require:** Vector database db | **Require:** Vector database db |
| 1: *// error attribution* | 1: *// error attribution* |
| 2: $\tilde{a}_t \leftarrow \arg\max_a \text{error}(a\|g, \tau)$ | 2: $\tilde{o}_t \leftarrow \arg\max_o \text{error}(o\|g, \tau)$ |
| 3: *// contrastive reflection* | 3: *// contrastive reflection* |
| 4: $\hat{o}_{t+1} \leftarrow \text{VLM}(g, \{o_0, a_0, ..., o_t, \tilde{a}_t\})$ | 4: $\hat{a}_t \leftarrow \text{VLM}(g, \{o_0, a_0, ..., \tilde{o}_t\})$ |
| 5: reflection $\leftarrow \text{VLM}(g, o_t, \tilde{a}_t, \{o_{t+1}, \hat{o}_{t+1}\})$ | 5: reflection $\leftarrow \text{VLM}(g, a_{t-1}, \tilde{o}_t, \{a_t, \hat{a}_t\})$ |
| 6: *// memorization* | 6: *// memorization* |
| 7: key $\leftarrow$ embedding($g, o_t$) | 7: key $\leftarrow$ embedding($g, o_t$) |
| 8: db $\leftarrow$ add(key, reflection) | 8: db $\leftarrow$ add(key, reflection) |
| 9: **return** | 9: **return** |

## A.2 R-MCTS PSEUDO CODE

**Overall Algorithm** We describe our R-MCTS agent in Algorithm 2. For clarity, we also abuse the notation $s$ to represent *observations* in this section, in order to comply with notations used in other works (Świechowski et al., 2022). We omitted details about action selection, expansion, simulation, and back-propagation as they are described in Appendix A.1. We present the main loop in R-MCTS

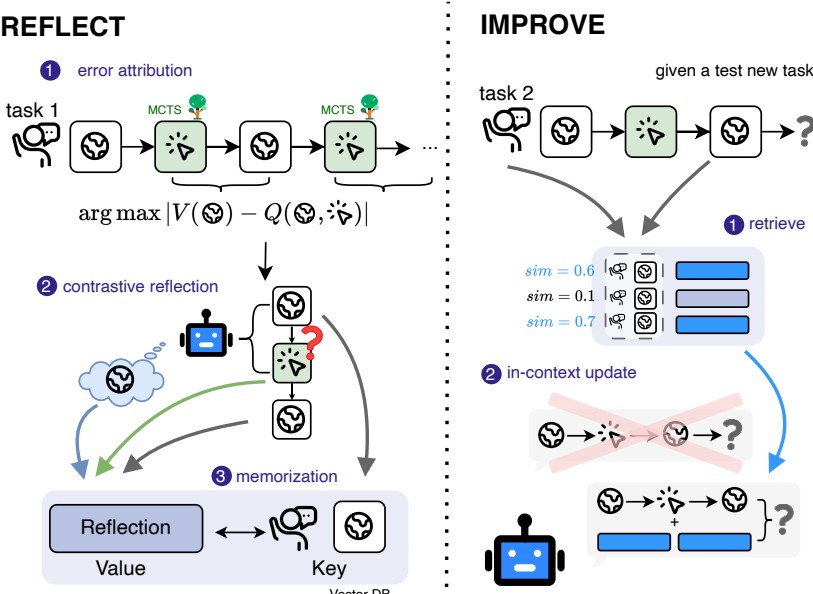

Figure 5: In-context reflection-improvement loop for an *R*-MCTS agent. For brevity, we only present policy reflection, as value reflection is similar. **Left**: after a complete episode, the agent 1) uses search tree statistics to select the most erroneous action $\tilde{a}_t$ from $\tau$ to reflect on; 2) prompts the VLM to generate a reflection by contrasting what it *expects* to achieve $\hat{o}_{t+1}$ and what *actually* happens $o_{t+1}$; 3) embeds the reflection in a vector database. **Right**: to generate an action in a new task, the agent 1) retrieves $m$ most relevant reflections from the database; 2) improves its policy (and value function) using in-context learning.

which consists of 1) an **inference loop** where the agent conducts tree search to find the best action for a given state; and 2) an **improvement loop** where the agent reflects on its past actions, and updates its policy and value function in-context using retrieval for future tasks. We detail the reflection and update process below.

**Contrastive Reflection** We present the pseudo code for both policy contrastive reflection and value contrastive reflection in Algorithm 3 and Algorithm 4, respectively, and illustrate the reflection-update process in Figure 5. This reflection process is repeated for $n_\pi$ times per trajectory for the top-$n_\pi$ erroneous actions during policy reflection, and $n_V$ times for value function reflection. We use $n_\pi = 3$ and $n_V = 1$ in our experiments. We use text-ada-003-small (OpenAI, 2024c) as the embedding model. For clarity, we repeat the error attribution equations in Section 3.1 below:

$$\text{error}_\pi(a_t|\tau) = |V(o_{t+1}) - Q(o_t, a_t)|, \quad \text{error}_V(o_t|\tau) = |V(o_t) - Q(o_{t-1}, a_{t-1})|. \quad (3)$$

**Improvement** Given a new task $g$ and trajectory $\tau = \{o_0, a_0, ..., o_t\}$, we retrieve the $m = 2$ most relevant reflections from the database, and use them to improve the policy and value function by appending them into the existing context (see Appendix E.2 for the prompt template). The retrieval process is done by computing the cosine similarity between the current task and observation embedding$(g, o_t)$ and the keys stored in the vector database. Since many tasks/webpages are unique, we also set a minimum similarity threshold of 0.25 to only use relevant reflections.

| Method | Task Affected by ID Error | Avg. Time per Task | Success Rate |
|---|---|---|---|
| Koh et al. (2024b) | 40.00% | 33.47 min | 45.00% |
| +Asyncio | 40.00% | 14.65 min | 35.00% |
| +Caching | 35.00% | 10.03 min | 40.00% |
| +Action Re-mapping | 10.00% | 10.10 min | 45.00% |

Table 6: Ablation studies of our modified browser implementation. Results are evaluated using SEARCH AGENT over 20 randomly sampled task in the Classifieds environment.

| Method | Model | Success Rate Before | Success Rate After |
|---|---|---|---|
| REACT | GPT-4o-mini | 10.67% | 14.67% |
| SEARCH AGENT | GPT-4o-mini | 18.67% | 22.67% |
| REACT | GPT-4o | 14.67% | 27.27% |
| SEARCH AGENT | GPT-4o | 26.67% | 38.57% |

Table 7: Performance comparison after updating task intents in VWA. We measure the task success before and after the updates for 75 randomly sampled tasks in the Classifieds environment.

## B ADDITIONAL DETAILS ABOUT VWA

### B.1 WEB BROWSER

Search-augmented agents require frequent backtracking to explore new actions. As a result, we find prior implementations of the *web browser* is inefficient and error prone: 1) all processing is done in a single thread sequentially; 2) every (backtracked) page is re-processed from scratch; and 3) can frequently cause action *element id mismatch* errors since backtracked page may not always be the same as before (e.g., *search history* is stored in the website's database and cannot be reset).

We thus modified the existing web browser used in Koh et al. (2024b), and used the same modified browser for *all methods* in our experiments. Our modifications include:

1. use `asyncio` to parallelize webpage processing (*Asyncio*);
2. implement a caching mechanism to store and retrieve SoM processing output when the same webpage is encountered (*Caching*);
3. when SoM element ids changed after backtracking, we implement a simple heuristic (word overlap of some metadata associated with the element) to re-map the VLM generated action id back to the correct element id (*Action Re-mapping*).

We present a quantitative study of these changes in Table 6.

### B.2 VWA EVALUATION

One common error we find with GPT-4o based agents (e.g., using REACT) is to directly return the (correct) item's name and description from (e.g.,) the search result page, while VWA evaluates these tasks by checking if the *final state's URL* is the desired item's URL. However, we believe this is due to ambiguously phrased task intent, such as "Find me the cheapest blue kayak on this site", and "Find the most expensive car from Virginia that is neon green", which does not explicitly mention/require navigating to the item's page to complete.

To this end, we updated these task instructions by appending the sentence: "Finish the task by navigating to that item's page and returning me the url link(s)" to these tasks for clarity. This change affected 93 instances in the Classifieds environment, 0 instance in the Reddit environment, and 34 instances in the Shopping environment. We present a quantitative study of this change in Table 7, and find that all methods/VLMs significantly improved by simply aligning the task intent with the evaluation metric. Unless otherwise specified, we use these updated task intents for all methods

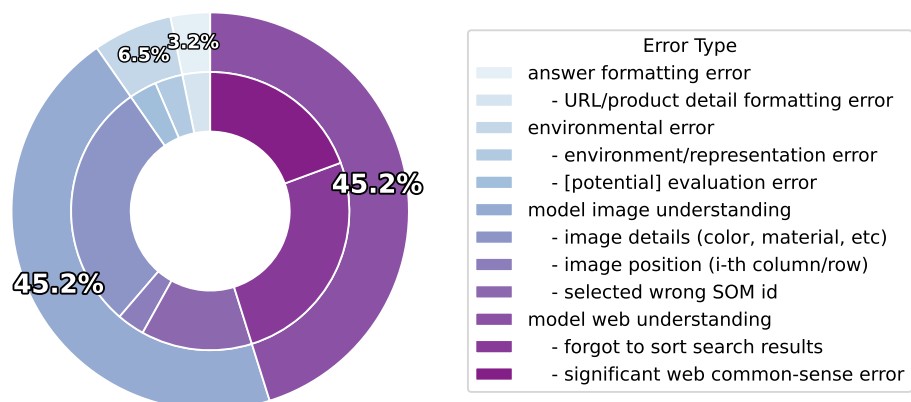

Figure 6: Error analysis for R-MCTSMAD after manually inspecting the search trees for 80 randomly sampled tasks from the Classfieds environment.

and experiments in this work. The comprehensive leaderboard for VisualWebArena can be found in Table 10.

## C    OSWORLD RESULTS

**Baselines**    For evaluations on OSWorld, we directly compare against existing best methods from the official leaderboard. This includes AGENT-S (Agashe et al., 2024), a prompt-based framework that augments agents with online web search and an experience-augmented hierarchical planning module; AGENTSTORE (Jia et al., 2024) a method that uses multiple (over 20) specialized agents as well as finetuning methods such as self-instruct to improve performance; and LEARN-BY-INTERACT (Anonymous, 2024), a data-centric framework using synthetic data (manually created based on related documentations and agent-environment interaction histories) to improve performance by either in-context learning or further training. In general, these methods rely on using additional resources such as online web search, task-specific models, or performs model training to improve performance.

| Method | Model | Training | OS Success | Office Success | Daily Success | Profess. Success | Workflow Success | Overall Success |
|---|---|---|---|---|---|---|---|---|
| REACT | GPT-4o | ✗ | 41.67% | 6.16% | 12.33% | 14.29% | 7.46% | 11.21% |
| AGENT-S | GPT-4o + Web | ✗ | 45.83% | 13.83% | 30.46% | 36.73% | 10.53% | 20.58% |
| AGENTSTORE(ICL) | Hybrid | ✗ | - | 1.71% | 32.05% | 22.44% | - | 13.55% |
| AGENTSTORE(FT) | Hybrid | ✓ | - | 15.38% | 37.18% | 12.20% | - | 17.34% |
| AGENTSTORE(AT) | Hybrid | ✓ | - | 28.20% | 37.18% | 23.93% | - | 23.85% |
| R-MCTS | GPT-4o | ✗ | 45.83% | 15.56% | 25.27% | 22.44% | 11.52% | 19.39% |

Table 8: Comparing different agent's performance on OSWorld, using *screenshot + accessibility tree* as input. For AGENT-S and AGENTSTORE, we use results reported by the respective authors. "Training" indicates whether the method finetuned the agent model. Results with gray backgrounds uses either additional resources (e.g., web search), models, or performs training.

**Results**    We summarize the results evaluated using "screenshot + accessibility tree" input modality in Table 8, and also using "accessibility tree" only in Table 9. We find that R-MCTS shows competitive performance compared other state-of-the-art methods, despite only using on a single model with a search algorithm.

| Method | Model | Training | OS Success | Office Success | Daily Success | Profess. Success | Workflow Success | Overall Success |
|---|---|---|---|---|---|---|---|---|
| REACT | GPT-4o | ✗ | 20.83% | 6.99% | 16.81% | 16.33% | 7.56% | 11.4% |
| LEARN-BY-INTERACT | Gemini-1.5-pro + RAG | ✗ | - | - | - | - | - | 10.3% |
| | Claude-3.5-sonnet + RAG | ✗ | - | - | - | - | - | 22.5% |
| R-MCTS | GPT-4o | ✗ | 37.50% | 10.41% | 21.82% | 24.49% | 10.87% | 16.6% |

Table 9: Comparing different agent's performance on OSWorld, using *accessibility tree* as input. Results with gray backgrounds uses either additional resources (e.g., RAG with a manually populated database) or different models.

| Model | Inputs | Success Rate (%) |
|---|---|---|
| EXACT (GPT-4o + R-MCTS$_{MAD}$) | SoM + Caption + Image | **33.74** |
| EXACT (GPT-4o + R-MCTS$_{SA}$) | SoM + Caption + Image | 32.53 |
| GPT-4o + MCTS | SoM + Caption + Image | 30.22 |
| GPT-4o + Search | SoM + Caption + Image | 26.40 |
| GPT-4o + ICAL | SoM + Caption + Image | 23.40 |
| GPT-4o | SoM + Caption + Image | 19.78 |
| Llama-3-70B + Search | AxTree + Caption | 16.70 |
| GPT-4V | SoM + Caption + Image | 16.37 |
| GPT-4 + BLIP-2-T5XL | AxTree + Caption + Image | 15.05 |
| GPT-4 | AxTree + Caption | 12.75 |
| Gemini-Pro-1.5 | SoM + Caption + Image | 11.98 |
| Llama-3-70B-Instruct + BLIP-2-T5XL | AxTree + Caption | 9.78 |
| GPT-4 | AXTree | 7.25 |
| Gemini-Flash-1.5 | SoM + Caption + Image | 6.59 |
| Gemini-Pro | SoM + Caption + Image | 6.04 |
| Gemini-Pro | SoM + Caption + Image | 5.71 |
| Gemini-Pro + BLIP-2-T5XL | AxTree + Caption | 3.85 |
| GPT-3.5 + BLIP-2-T5XL | AxTree + Caption | 2.97 |
| GPT-3.5 + LLaVa-7B | AxTree + Caption | 2.75 |
| GPT-3.5 | AXTree | 2.20 |
| Gemini-Pro | AXTree | 2.20 |
| Mixtral-8x7b + BLIP-2-T5XL | AxTree + Caption | 1.87 |
| Mixtral-8x7b | AXTree | 1.76 |
| Llama-2-70B | AXTree | 1.10 |
| IDEFICS-80B-Instruct | SoM + Caption + Image | 0.99 |
| IDEFICS-80B-Instruct | AXTree + Caption + Image | 0.77 |
| Llama-2-70B + BLIP-2-T5XL | AxTree + Caption | 0.66 |
| CogVLM | SoM + Caption + Image | 0.33 |
| CogVLM | AXTree + Caption + Image | 0.33 |

Table 10: VisualWebArena Leaderboard

# D ADDITIONAL ANALYSIS DETAILS

## D.1 ERROR ANNOTATION SCHEME

We label errors made by our R-MCTS agent into 4 primary categories (answer formatting error, environmental error, model text understanding, and model image understanding). We find these categories covered most of the error cases. We further sub-divided these categories into 8 sub-error types for a more detailed analysis, and present the result in Figure 6. We detail the error types below.

**Answer Formatting**  Answer formatting errors include: 1) returning the item's name/description instead of its URL used for evaluation, or failing to format a range of prices into the instructed format (*URL/product detail formatting error*).

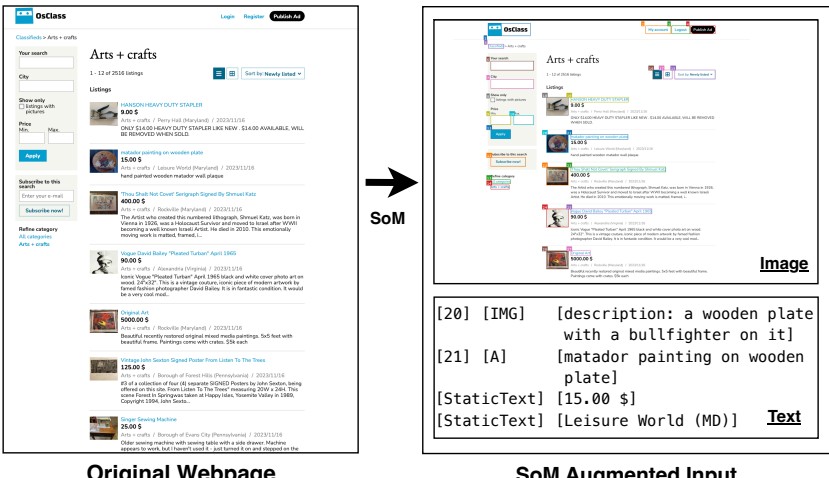

Figure 7: Example Set-of-Mark augmented webpage used as inputs for VLM agents.

**Environmental Errors**    Environmental errors include: 1) incompatibility issues with certain website functionality (e.g., dropdown menu) with SoM augmentation (*environment/representation error*); and 2) ambiguous task intents leading to more than one potentially correct answer, one of which the model returned (*potential evaluation error*).

**Model Image Understanding**    Model image understanding errors include: 1) failing to correctly identify certain properties (e.g., color, pattern, material, etc.) from an image (*image details*); 2) failing to identify product on the i-th row and/or j-th column (*image position*); and 3) generating an action with element ID not present on the current observation (*selected wrong SoM id*).

**Model Web Understanding**    Model web understanding errors include: 1) forgetting to sort search results and misunderstands the most expensive/cheapest item on the current page is also the most expensive/cheapest item on the entire website (*forgot to sort search results*); and 2) other errors potentially caused by a lack web common-sense knowledge, such as trying to filter for price ranges by typing in the *search bar* instead of using price filters, and returning related items on other pages when specifically instructed to find it on the *current* page (*significant web common-sense error*).

## E    ADDITIONAL PROMPTING DETAILS

### E.1    SET-OF-MARK AUGMENTATION

We follow Koh et al. (2024a;b) and represent visual webpages using Set-of-Mark (SoM) (Yang et al., 2023b) augmentation. SoM augments a webpage by 1) creating bounding boxes around each interactive element on the page; and 2) adding a unique identifier to each element. We present an example in Figure 7.

### E.2    PROMPTS FOR R-MCTS AGENT

**Prompts for Contrastive Reflection**    We present the prompts to generate an expected observation given a previous state an action in Table 11. We then generate a reflection using use the prompts in Table 12.

**Prompts for Improvement**    We present the prompts used to format the relevant reflections retrieved from the vector database back into model's current context in Table 13.

| Role | Prompt |
|------|--------|
| System | You are an autonomous intelligent agent tasked with navigating a web browser. You will be given web-based tasks. These tasks will be accomplished through the use of specific actions you can issue.
Here's the information you'll have: (... some content omitted) |
| User | OBSERVATION:
[] [StaticText] [What are you looking for today?]
[5] [INPUT] []
[6] [SELECT] [Select a category]
[7] [BUTTON] [Search]
[] [StaticText] [Latest Listings]
[] [StaticText] [Atlas Powered Audio System w/ Tripod 150.00 $ Music instruments Borough of Red Lion (Pennsylvania) 2023/11/16]
(... some content omitted)
OBJECTIVE: Find me the cheapest red Toyota. It should be between $3000 to $6000. Finish the task by navigating to that item's page and returning me the url link(s).
IMAGES: (1) current page screenshot
image_url: data:image/png;base64,iVBORw0KGgoAAAANSUh... |
| Assistant | Based on the objective, I need to find the cheapest red Toyota priced between $3000 to $6000. (... some content omitted)
In summary, the next action I will perform is "'type [5] [red Toyota] [1]'" |
| User | What do you expect to happen after taking this action? Briefly describe what you think will appear on the webpage after performing the action. |
| Assistant | After taking this action, I expect the webpage to display search results displaying red Toyotas, (... some content omitted) |

Table 11: Prompts used to generate the expected state transition. *image_url* is the base64 encoded SoM augmented web screenshot. Multimodal input in highlighted in orange. Generated response is highlighted in blue.

| Role | Prompt |
|------|--------|
| System | You are an autonomous intelligent agent tasked with navigating a web browser. You will be given web-based tasks. These tasks will be accomplished through the use of specific actions you can issue. Here's the information you'll have: (... some content omitted) |
| User | OBSERVATION: 
 [] [StaticText] [What are you looking for today?] 
 [5] [INPUT] [] 
 [6] [SELECT] [Select a category] 
 [7] [BUTTON] [Search] 
 [] [StaticText] [Latest Listings] 
 [] [StaticText] [Atlas Powered Audio System w/ Tripod 150.00 $ Music instruments Borough of Red Lion (Pennsylvania) 2023/11/16] 
 (... some content omitted) 
 OBJECTIVE: Find me the cheapest red Toyota. It should be between $3000 to $6000. Finish the task by navigating to that item's page and returning me the url link(s). 
 IMAGES: (1) current page screenshot 
 image_url: data:image/png;base64,iVBORw0KGgoAAAANSUh... |
| Assistant | Based on the objective, I need to find the cheapest red Toyota priced between $3000 to $6000. (... some content omitted) 
 In summary, the next action I will perform is "'type [5] [red Toyota] [1]'" 

 Expectation: After taking this action, I expect the webpage to display search results displaying red Toyotas, (... some content omitted) |
| User | NEXT OBSERVATION: 
 [596] RootWebArea 'red Toyota - Classifieds' focused: True 
 [630] link 'Classifieds, description: a blue and black logo...' 
 [663] img 'Classifieds, description: a blue and black logo...' 
 [638] link 'My account' 
 [639] link 'Logout' 
 [640] link 'Publish Ad' 
 [647] link 'Classifieds' 
 (... some content omitted) 
 IMAGES: (1) next page screenshot 
 Image URL: data:image/png;base64,iVBORw0KGgoAAA... 

 Is this webpage what you expected? If not, can you conclude anything special about navigating on this website? If you faced the same situation again, what would you do differently at a high level? Do NOT propose any specific actions/answers. 
 Keep your response within 100 words. |
| Assistant | This webpage does not display the expected car listings but rather includes various (... some content ommitted) |

Table 12: Prompt used to generate contrastive reflections given an expected observation highlighted in teal. *image_url* is the base64 encoded SoM augmented web screenshot. Multimodal input in highlighted in orange. Generated response is highlighted in blue.

| Role | Prompt |
|------|--------|
| System: | You are an autonomous intelligent agent tasked with navigating a web browser. You will be given web-based tasks. These tasks will be accomplished through the use of specific actions you can issue.
Here's the information you'll have: (... some content omitted) |
| User | REFLECTIONS: here are some relevant reflections from other tasks. Note that these reflections may not be directly related to the new task below, but they may provide some useful insights.

OBJECTIVE (1): I recall seeing this exact item (... some content omitted)
ATTEMPTED ACTION (1): Let's think step-by-step. The objective is to find the most recent post of the item shown in the image, which is a personal watercraft. To find this, I will search for "watercraft" in the search box. (... some content omitted)
In summary, the next action I will perform is "'type [5] [watercraft] [1]'"
REFLECTION (1): This webpage (... some content omitted). However, it did not directly show the item from the provided image. The search term might have been too broad. (...some content omitted)
#####
OBJECTIVE (2): Tell me the name of the lister with (... some content omitted)
ATTEMPTED ACTION (2): Let's think step-by-step. The objective is to find the name of the lister with the most expensive green vehicle from West Virginia. (...some content ommitted)
In summary, the next action I will perform is "'click [60]'"
REFLECTION (2): Yes, the webpage is what I expected, displaying listings specifically from West Virginia. However, it appears that filtering by color (green) is not directly possible. (... some content omitted)

!IMPORTANT! Below is the task you need to solve. Please read the user's intent and input images (if any) carefully.
OBSERVATION:
[1] [IMG] [Classifieds, description: a blue and black logo with the words ohm, url: http://coffee.cs.columbia.edu:57981...]
[2] [A] [My account]
[3] [A] [Logout]
[4] [A] [Publish Ad]
[] [StaticText] [What are you looking for today?]
(... some content omitted)
URL: http://classifieds.com/
OBJECTIVE: How many yellow or blue motorcycles in total were posted on 25th October 2023?
PREVIOUS ACTION: None
NOTE: Remember that you should consider user's intent and reflections (if applicable) to better plan the next action.
IMAGES: (1) current page screenshot
Image URL: data:image/png;base64,iVBORw0KGgoAAAAN... |
| Assistant | Let us think step by step. The objective is to find the total number of yellow or blue motorcycles posted on 25th October 2023. From the reflections (... some content omitted) |

Table 13: Prompt used to generate an action given retrieved reflections, which is highlighted in teal. *image_url* is the base64 encoded SoM augmented web screenshot. Multimodal input in highlighted in orange. Generated response is highlighted in blue. For brevity, we present prompts for trajectories with only one action executed so far.

| Role | Prompt |
|---|---|
| System | You are an autonomous intelligent agent tasked with navigating a web browser. You will be given web-based tasks. These tasks will be accomplished through the use of specific actions you can issue.
Here's the information you'll have: (... some content omitted) |
| User | User Intent: Find me the cheapest red Toyota. It should be between $3000 to $6000. Finish the task by navigating to that item's page and returning me the url link(s).
IMAGES: (1) start page screenshot
Image URL: data:image/png;base64,iVBORw0KGgoAAAAN... |
| Assistant | type [5] [red Toyota] where [5] is [INPUT] element with content [] |
| User | Last page URL=http://coffee.cs.columbia.edu:57981/index.php?page=...
IMAGES: (1) last page screenshot.
Image URL: data:image/png;base64,iVBORw0KGgoAAA...

Now, it is your turn to evaluate the success/failure of the agent's execution so far. (... some content omitted)
And your goal is to find the most convincing evidence why the agent is NOT on the right track to complete the task. (... some content omitted)
Keep your response within 100 words. |
| Assistant | The agent has not successfully completed the task. The user's intent was to find the cheapest red Toyota priced between $3000 and $6000. The search results include vehicles that are outside the price range and have the wrong color (... some content omitted) |

Table 14: Prompts used to generate opposing arguments for value estimation. *image_url* is the base64 encoded SoM augmented web screenshot. Multimodal input in highlighted in orange. Generated response is highlighted in blue. For brevity, we present prompts for trajectories with only one action executed so far.

| Role | Prompt |
|------|--------|
| System | You are an autonomous intelligent agent tasked with navigating a web browser. You will be given web-based tasks. These tasks will be accomplished through the use of specific actions you can issue.
Here's the information you'll have: (... some content omitted) |
| User | User Intent: Find me the cheapest red Toyota. It should be between \$3000 to \$6000. Finish the task by navigating to that item's page and returning me the url link(s).
IMAGES: (1) start page screenshot
Image URL: data:image/png;base64,iVBORw0KGgoAAAAN... |
| Assistant | type [5] [red Toyota] where [5] is [INPUT] element with content [] |
| User | Last page URL=http://coffee.cs.columbia.edu:57981/index.php?page=...
IMAGES: (1) last page screenshot.
Image URL: data:image/png;base64,iVBORw0KGgoAAA...

Now, it is your turn to evaluate whether the agent's actions so far are successfully aligned with the user's intent. (... some content omitted)

To better verify if user's intent is fulfilled correctly, you may find the following opinions helpful:
Reasons why the agent is NOT on the right track:
The agent has not successfully completed the task. The user's intent was to find the cheapest red Toyota priced between \$3000 and \$6000. The search results include vehicles that are outside the price range and have the wrong color (... some content omitted)
Reasons why the agent is on the right track:
The agent has successfully navigated to the search results page for "red Toyota". Specifically, the "2007 Toyota Yaris" at \$3000 and the "2007 Toyota Corolla" at \$6500 are relevant. (...some content omitted)

Note that these opinions may or may NOT be correct. You should make your own judgment based on the user's intent, the observations, and the agent's executed actions so far.

To make a final decision, choose one of the following status codes and provide your thought process.
STATUS CODES:
A. The agent's last action is "`stop`" and it contains the correct answer (...some content omitted)
B. The agent is very close to finishing the task (...some content omitted)
C. The agent still needs a few more actions (...some content omitted)
D. (...some content omitted)
E. (...some content omitted) |
| Assistant | Thoughts: The agent has successfully navigated to the results page for "red Toyota". However, it hasn't identified the cheapest (...some content omitted)
STATUS CODE: C |

Table 15: Prompts used to generate the final judgement for value estimation, given opposing arguments highlighted in teal. *image_url* is the base64 encoded SoM augmented web screenshot. Multimodal input in highlighted in orange. Generated response is highlighted in blue. For brevity, we present prompts for trajectories with only one action executed so far.

**Prompts for Multi-Agent-Debate** We present the prompts to perform multi-agent-debate in Table 14, and Table 15. For simplicity, we present prompts to generate opposing arguments for value estimation Table 14, since prompts to generate supporting arguments is highly similar.

