# OpenReview forum: "ExACT: Teaching AI Agents to Explore with Reflective-MCTS and Exploratory Learning"
_ICLR.cc/2025/Conference — ICLR 2025 Poster_

### Official Review · Reviewer_SSY4 · 2024-10-24

**Soundness:** 2
**Presentation:** 2
**Contribution:** 2
**Rating:** 6
**Confidence:** 2

**Summary:**

The paper presents a variation of MCTS that exploit the reasoning capabilities of VLMs and that can also be fine tuned.

**Strengths:**

1. Incorporating VLMs reasoning capabilities into MCTS is a novel idea with great potential.
2. The proposed methodology has been extensively evaluated on a challenging benchmark and significantly outperforms the baselines.
3. Improvements of the benchmark with optimizations that will useful for future works.

**Weaknesses:**

1. The presentation of the paper makes it hard to follow. For example, in the second paragraph of Section 3.1 the authors say that the mult-agent debate state estimation was introduced in Section 2.3 but in that section they say that the methodology will be described in the next section.
2. The methodology seems to be specifically tailored for the web navigation task, and it is potentially unlikely to generalize to different tasks. A clearer explanation and a better contextualization would improve the clarity of the paper.

**Questions:**

1. Why is the methodology only evaluated in web? Is it because it requires access to the model of the environment to run the MCTS?
2. Can you please better explain how the simulation step carried out in the algorithm? In vanilla MCTS, it is carried out with random rollout. Is it different in the proposed methodology?
2. In Table 2, you show the results for several search methodology with a different amount of tokens? Would it be possible to have a comparison with a fixed computational budget? I think it would be a more fair comparison with the baselines.
3. I wasn't able to understand what the values in Training column from Table 3 means, can you clarify?

**Details Of Ethics Concerns:**

No concerns.

---

> ### Author Response · Authors · 2024-11-17
> **Response 1/2**
>
> Thank you for providing these questions and suggestions!
>
> ---
>
> > The presentation of the paper makes it hard to follow. For example, in the second paragraph of Section 3.1 the authors say that the mult-agent-debate state estimation was introduced in Section 2.3 but in that section they say that the methodology will be described in the next section.
>
> Sorry for the confusion. We believe there is a typo on L175, where it stated “multi-agent-debate (Section 2.3)” instead of “multi-agent-debate”. We overlooked this mistake as we have re-organized this paper several times trying to improve readability. We will fix this in our final manuscript.
>
> We clarify that our work focuses on how to effectively scale test-time compute to improve VLM agents, and efficiently transfer knowledge acquired from search back to the model to enhance its reasoning and planning abilities. We achieve this by introducing a new search algorithm R-MCTS (Section 3.1), and new training methods such as Tree-Traversal SFT (Section 3.2). We then experimented on VisualWebArena, and presented our R-MCTS results in Section 4.1, training results in Section 4.2, and an additional scaling experiment in Section 4.3. Finally, we provided numerous analyses in Section 5, such as an ablation study (Section 5.2) and a qualitative analysis of R-MCTS (Section 5.3).
>
>
> > The methodology seems to be specifically tailored for the web navigation task, and it is potentially unlikely to generalize to different tasks.
>
> VisualWebArena (and WebArena) are challenging yet realistic benchmarks to evaluate autonomous multimodal agents' performance on executing tasks on a computer. *Practically, R-MCTS agent only requires a visual or textual representation of the environment state (e.g., a screenshot) and outputs actions such as clicking, typing, scrolling, etc.* All methods do **not** have access to any unrealistic web resources, such as the backend web implementation. This makes it easy to transfer between many tasks and domains. To demonstrate this, we additionally present results from evaluating on the GitLab domain from [3]:
>
> | Method          | Search      | Value | GitLab (Tokens) | GitLab (Success) |
> |-----------------|-------------|-------|-----------------|------------------|
> | ReACT           | -           | -     | 1x              | 11.3%            |
> | ToT             | BFS         | SA    | 4.6x            | 14.1%            |
> | ToT             | DFS         | SA    | 4.3x            | 14.1%            |
> | Search Agent    | Best-First  | SA    | 3.0x            | 13.8%            |
> | MCTS            | MCTS        | SA    | 5.7x            | 19.9%            |
> | R-MCTS (ours)   | MCTS        | SA    | 6.0x            | 20.9%            |
> | R-MCTS (ours)   | MCTS        | MAD   | 8.8x            | 23.5%            |
>
> We believe this is a generic and extensible setup and can be extended to other benchmarks such as Operating System tasks [1] and Android tasks [2].
>
> We are sorry if we misunderstood your original question. If the above does not resolve your concern, could you please elaborate more why it is potentially hard to generalize to different tasks?
>
>
>
> > Why is the methodology only evaluated on the web? Is it because it requires access to the model of the environment to run the MCTS?
>
> We evaluated VisualWebArena as it offers a suite of diverse, realistic, and reproducible environments including shopping (Shopping), social forums (Reddit), and content management (CMS). Practically, our agent only requires access to a visual or textual representation of the environment state (e.g., a screenshot), and *is independent of the backend web implementation*. We will clarify this in our section 2.3.
>
>
> > Can you please better explain how the simulation step was carried out in the algorithm? In vanilla MCTS, it is carried out with random rollout. Is it different in the proposed methodology?
>
> Sorry for the confusion. Our rollout policy is based on the current agent, and simulation is based on executing the action in an actual browser. We will clarify this in our Appendix A.1.

---

> ### Author Response · Authors · 2024-11-17
> **Response 2/2**
>
> > In Table 2, you show the results for several search methods with a different amount of tokens? Would it be possible to have a comparison with a fixed computational budget?
>
> We clarify that we did not intentionally increase the budget allocated to R-MCTS. All agents (ToT, Search Agent, MCTS, and R-MCTS) are run under the same search budget and hyperparameters such as width and breadth factor (please also see our general response GR-1). However, the greedy best-first nature of algorithms such as ToT BFS/DFS and Search Agent makes them often terminate early as they keep expanding states with high *estimated* value.
>
> Under equal token usage, we show that:
> - Best-of-N ReACT using MAD underperforms R-MCTS under all token usages as shown in the table in GR-2.
> - R-MCTS achieves 35.0% on Classifieds with 4.0x ReACT token usage (from Figure 1 left), whereas Search Agent achieves only 33.8% with 4.2x ReACT token usage (from Table 2).
>
>
> > I wasn't able to understand what the values in the Training column from Table 3 means, can you clarify?
>
> Sorry for the confusion. The “Training” column indicates whether the VLM used has undergone training mentioned in Section 3.2, and if so, how many trajectories are used. For example, the first two rows in Table 3 directly prompt OpenAI’s GPT-4o-mini and o1-mini, respectively, to complete tasks on VWA. In contrast, the last two rows prompt GPT-4o that has undergone BiT SFT and TrT SFT training with 65 and 35 trajectories, respectively.
>
> We will clarify this in the caption in Table 3.
>
> ---
>
> **References**
>
> [1] Xie, Tianbao et al. “OSWorld: Benchmarking Multimodal Agents for Open-Ended Tasks in Real Computer Environments.” ArXiv abs/2404.07972 (2024): n. pag.
>
> [2] Rawles, Christopher et al. “Android in the Wild: A Large-Scale Dataset for Android Device Control.” ArXiv abs/2307.10088 (2023): n. Pag.
>
> [3] Zhou, Shuyan et al. “WebArena: A Realistic Web Environment for Building Autonomous Agents.” ArXiv abs/2307.13854 (2023): n. pag.

---

> ### Author Response · Authors · 2024-11-19
>
> Thank you again for these questions and feedback! Please let us know at your earliest convenience if you have any further questions or concerns, or would like to us conduct any additional experiments.

---

> > ### Author Response · Authors · 2024-11-20
> >
> > Thank you again for your feedback and thoughtful questions! As we are now halfway through the rebuttal period, I’m following up again to kindly ask if you have any further questions or concerns regarding our responses. If so, please feel free to post them and we will try to answer them as soon as possible!

---

> ### Comment · Reviewer_SSY4 · 2024-11-24
>
> Thank you for the clarifications. I now understand the methodology better and I increase the score to 6.

---

> > ### Author Response · Authors · 2024-11-24
> >
> > Thank you for reading through our responses and increasing the score! We will provide clarifications on our experimental setup and fix the typos in our final manuscript.

---

### Official Review · Reviewer_XTzE · 2024-10-27

**Soundness:** 2
**Presentation:** 3
**Contribution:** 2
**Rating:** 5
**Confidence:** 4

**Summary:**

The authors propose to improve autonomous AI agents with reflective tree search and self-learning. The authors introduce Reflective Monte Carlo Tree Search (R-MCTS), a test-time algorithm to explore decision space on the fly, with contrastive reflection and multi-agent debate. The authors fine-tune GPT-4o through self-learning, using tree traversals generated by R-MCTS, without any human-provided labels. The authors conduct experiments with VisualWebArena benchmark.

**Strengths:**

Reflective Monte Carlo Tree Search (R-MCTS) for test-time exploration of decision space
fine-tuning GPT-4o through self-learning

**Weaknesses:**

A fundamental issue is:
Given that no LLMs are perfect, so that the data generated  and the evaluations conducted by LLMs are likely not fully reliable, although the algorithm may improve the performance.
It is desirable or a must to collect reliable training data and feedback for a healthy learning system.

**Questions:**

Why it is valid to
self-generate data from an imperfect LLM,
self-reflect with an imperfect LLM, and
fine-tune an imperfect LLM from data generated by itself?
Is this based on some sound principle?
Although the algorithm may improve the performance, what is the drawbacks and limitations of a learning system trained with data not fully reliable?

---

> ### Author Response · Authors · 2024-11-17
> **Response**
>
> Thank you for providing these questions and comments!
>
> ---
>
> > Why is it valid to self-generate data from an imperfect LLM, self-reflect with an imperfect LLM, and fine-tune an imperfect LLM from data generated by itself? Is this based on some sound principle?
>
> Thank you for the question. Self-improvement methods using model generated data are highly common [1,2,3,4,5]. This is because 1) evaluating a solution/trajectory is typically easier than generating it; and 2) training an LLM/VLM with data of higher performance (albeit noisy) than itself is alike RLHF [6,7] which uses data generated by the model itself judged by an imperfect reward model. Hence, many work in self-improvement explores methods to obtain high-quality data using LLM/VLMs, despite their imperfections.
>
> We will add this in our background section.
>
>
> > Given that no LLMs are perfect, so that the data generated and the evaluations conducted by LLMs are likely not fully reliable, although the algorithm may improve the performance.
>
> Self-improvement methods [1,2,3,4,5] using data generated by LLMs have shown improved performance in areas such as math, coding, creative writing, and more. Similar to these prior work, we trained GPT-4o in agentic tasks using higher-performance data (i.e., from R-MCTS) than the model itself (i.e., ReACT), and showed that it can match over 85% of R-MCTS’s performance while reducing token usage by upto 4x (see Section 4.2).
>
>
> > What are the drawbacks and limitations of a learning system trained with data not fully reliable?
>
> As data collected from self-improvement methods contains noises, the data collection and training loop will typically plateau after a few iterations. However, since our method uses on-policy data generated by the model itself, we believe it is also unlikely for the model to “unlearn” skills that it already possesses and significantly worsen performance.
>
> We will add this consideration in our limitation section.

---

> > ### Comment · Reviewer_XTzE · 2024-11-24
> > **Self-improvement based on a fixed, imperfect LLM is not a valid approach.**
> >
> > Thanks authors for the explanation.
> >
> > "Self-improvement methods using model generated data are highly common [1,2,3,4,5]."
> >
> > Having many reference does not make it convincing.
> > The fundamental issue is: Where is the reliable information?
> > LLMs are not perfect, so that they can not guarantee reliable information.
> > Self-improvement based on a fixed, imperfect LLM is not a valid approach.
> >
> > I will keep my assessment.

---

> ### Author Response · Authors · 2024-11-17
> **References**
>
> References:
>
>
> [1] Jiaxin Huang, Shixiang Gu, Le Hou, Yuexin Wu, Xuezhi Wang, Hongkun Yu, and Jiawei Han. 2023. Large Language Models Can Self-Improve. In Proceedings of the 2023 Conference on Empirical Methods in Natural Language Processing, pages 1051–1068, Singapore. Association for Computational Linguistics.
>
> [2] Gulcehre, Caglar et al. “Reinforced Self-Training (ReST) for Language Modeling.” ArXiv abs/2308.08998 (2023): n. pag.
>
> [3] Aksitov, Renat et al. “ReST meets ReAct: Self-Improvement for Multi-Step Reasoning LLM Agent.” ArXiv abs/2312.10003 (2023): n. Pag.
>
> [4] Yu, Xiao et al. “Teaching Language Models to Self-Improve through Interactive Demonstrations.” North American Chapter of the Association for Computational Linguistics (2023).
>
> [5] Hu, Chi et al. “Teaching Language Models to Self-Improve by Learning from Language Feedback.” Annual Meeting of the Association for Computational Linguistics (2024).
>
> [6] Ouyang, Long et al. “Training language models to follow instructions with human feedback.” ArXiv abs/2203.02155 (2022): n. pag.
>
> [7] Sun, Zhiqing et al. “SALMON: Self-Alignment with Instructable Reward Models.” International Conference on Learning Representations (2023).

---

> ### Author Response · Authors · 2024-11-19
>
> Thank you again for these questions and feedback! Please let us know at your earliest convenience if you have any further questions or concerns, or would like to us conduct any additional experiments.

---

> > ### Author Response · Authors · 2024-11-20
> >
> > Thank you again for your feedback and thoughtful questions! As we are now halfway through the rebuttal period, I’m following up again to kindly ask if you have any further questions or concerns regarding our responses. If so, please feel free to post them and we will try to answer them as soon as possible!

---

> ### Author Response · Authors · 2024-11-24
>
> Thank you for your reply.
>
> > Having many reference does not make it convincing... Self-improvement based on a fixed, imperfect LLM is not a valid approach.
>
> Sorry if we misunderstood your statement, but are you claiming that all these prior work in model self-improvement [1,2,3,4,5] as well as their respective prior/follow up work are not valid?
>
> We note that [1,2,3,4,5] are *only very few examples* we cited from the area of self-improvement research, which studies how to use a LLM/VLM to improve its own generation without additional human resources/signals. For more prior work in this area, we refer to papers that cites or follows up [1], [8] or [9], which we believe are one of the first papers that present convincing results in LLM self-improvement.
>
> > The fundamental issue is: Where is the reliable information? LLMs are not perfect, so that they can not guarantee reliable information
>
> On a high level, we note that this loop of "generating improved data with an LLM" and "training the LLM with these data" is also common in alignment research. For example, RLHF [6] and its numerous follow up work trains an LLM using its *own generation judged by a learned reward model*, which is often also an LLM.
>
> ---
> More References
>
> [8] Madaan, Aman et al. “Self-Refine: Iterative Refinement with Self-Feedback.” ArXiv abs/2303.17651 (2023): n. pag.
>
> [9] Bai, Yuntao, et al. "Constitutional ai: Harmlessness from ai feedback." arXiv preprint arXiv:2212.08073 (2022).

---

> > ### Comment · Reviewer_XTzE · 2024-11-24
> > **Self-improvement based on a fixed, imperfect LLM is not a valid approach.**
> >
> > Self-improvement based on a fixed, imperfect LLM is not a valid approach.
> >
> > One exception is: There is a verification stage for un-reliable signals from imperfect LLMs.
> >
> > It is fundamental, regardless of how many previous published papers.
> >
> >
> > A recent blog by Anthropic talks about adding error bars to statistical results.
> >
> > Most LLMs papers do not have error bars or confidence intervals, including your paper and the ReAct paper.
> >
> > It is such a basic research issue.
> >
> > Without error bars or confidence intervals, why trust papers like ReAct?
> > (Lacking of reliable signals is one more issue.)

---

> > > ### Author Response · Authors · 2024-11-25
> > >
> > > Thank you for your additional response.
> > >
> > > ---
> > >
> > > > Self-improvement based on a fixed, imperfect LLM is not a valid approach. One exception is: There is a verification stage for unreliable signals from imperfect LLMs.
> > >
> > > It is unclear what verification stage you are referring to. Is it a verification from LLM or from some external programs that can mitigate unreliability? Could you elaborate in more details about the “verification stage”? In the area of self-improvement research, the verification from LLM we reckon are not 100% accurate.
> > >
> > > Besides the assertions, we would appreciate if you could provide concrete evidences, rationales behind the assertions, and constructive and actionable suggestions.
> > >
> > >
> > > > A recent blog by Anthropic talks about adding error bars to statistical results… Without error bars or confidence intervals, why trust papers like ReAct [as well as our work]?
> > >
> > > As to your question “without error bars or confidence intervals, why trust papers like ReAct”, we refrain from commenting on it. It is worth noting, however, that ReAct has been accepted by ICLR 2024 and cited for 1700+ times, and the absence of error bars in such papers (including GPT-4/Claude/LLama report) does not preclude their significant impact.
> > >
> > > We follow ReAct [11] and ToT [12] and directly present success rate without error bars, since R-MCTS_{MAD) outperforms prior best [13] by 27.7% relatively, which is substantial. But we appreciate your emphasis on the importance of error bars or confidence intervals to strengthen the reliability of experimental results. We adopt the bootstrap method by sampling 90% of the test data 100 times to calculate the mean and standard deviation for the classifieds domain, and present the results below. We observe that at 95% confidence interval, the R-MCTS (ours) method demonstrates superior performance compared to ReACT, ToT (both BFS and DFS variants), Search Agent, and MCTS.
> > >
> > > | Method             | Search       | Value | Success (Std)    |
> > > |--------------------|--------------|-------|------------------|
> > > | ReACT             | -            | -     | 28.6% (±0.9%)    |
> > > | ToT               | BFS          | SA    | 30.7% (±0.9%)    |
> > > | ToT               | DFS          | SA    | 30.3% (±0.7%)    |
> > > | Search Agent      | Best-First   | SA    | 33.8% (±1.0%)    |
> > > | MCTS              | MCTS         | SA    | 37.6% (±1.1%)    |
> > > | R-MCTS (ours)     | MCTS         | SA    | 40.2% (±0.7%) |
> > > | R-MCTS (ours)     | MCTS         | MAD   | **41.0% (±0.9%)** |
> > >
> > >
> > > As to the blog post, are you referring to [10] that appears online on Nov 19 2024? Please provide concrete references and rationales supporting your claims such that we can make actionable changes.
> > >
> > >
> > > ---
> > >
> > > Additional References:
> > >
> > >
> > > [10] https://www.anthropic.com/research/statistical-approach-to-model-evals
> > >
> > > [11] Yao, Shunyu, et al. "React: Synergizing reasoning and acting in language models." arXiv preprint arXiv:2210.03629 (2022).
> > >
> > > [12] Yao, Shunyu, et al. "Tree of thoughts: Deliberate problem solving with large language models." Advances in Neural Information Processing Systems 36 (2024).
> > >
> > > [13] Koh, Jing Yu, et al. "Tree search for language model agents." arXiv preprint arXiv:2407.01476 (2024).

---

> > > > ### Comment · Reviewer_XTzE · 2024-11-25
> > > > **the "reliable signal" issue is fundamental**
> > > >
> > > > External verifier, e.g., maths solvers like Lean.
> > > > LLMs are not perfect, so they are not qualified as verifiers.
> > > >
> > > > Yes, it is https://www.anthropic.com/research/statistical-approach-to-model-evals,
> > > > actually the arXiv paper https://arxiv.org/abs/2411.00640
> > > >
> > > > I am surprised with the following comment:
> > > > 1) being accepted by a top AI conference and the # of citations are so important, and
> > > > 2) the disdain with error bars, comparing with top venue publication / citation.
> > > >
> > > > "that ReAct has been accepted by ICLR 2024 and cited for 1700+ times, and the absence of error bars in such papers (including GPT-4/Claude/LLama report) does not preclude their significant impact"
> > > >
> > > > Thanks for providing CI.
> > > > However, the "reliable signal" issue is fundamental; ReAct being one of early reference, and the problem is treating an imperfect LLM as an oracle.

---

> ### Author Response · Authors · 2024-11-25
>
> Thanks for acknowledging our responses on providing the CI. We hope we have at least addressed your concern on trustiness of our method!
>
> ---
>
> > I am surprised with the following comment: being accepted by a top AI conference and the # citations are so important, and the disdain with error bars, compared to top venue publication / citation.
>
> Thank you for referencing our previous responses. To provide additional context, consider the following statement we made:
>
> ```
> As to your question “without error bars or confidence intervals, why trust papers like ReAct”, we refrain from commenting on it. It is worth noting, however, that ReAct has been accepted by ICLR 2024 and cited for 1700+ times, and the absence of error bars in such papers (including GPT-4/Claude/LLama report) does not preclude their significant impact.
>
> ```
> **We respectfully disagree with your perception of our stance**. We trust in the discretion of ICLR 2024 and the broader research community, as well as in the validity of the results presented in the ReAct paper.
> More importantly, we have already acknowledged and agreed with your emphasis on the importance of including error bars and CI. We completely concur that their inclusion strengthens the reliability of experiment results. This is why we provided error bars and CI in our work to address this very point.
>
> **We are unsure how you derived the impression that we "disdain error bars" from any of our prior responses**. If there is a specific statement or tone that led to this misunderstanding, we would like to make corrections.
>
> >  the problem is treating an imperfect LLM as an oracle.
>
> Regarding your main concern of “reliable signal” and the raised problem “treating an imperfect LLM as an oracle": we **never stated** or conveyed in our paper that we treat the value function (an imperfect LLM) as an oracle one. We appreciate it if you can provide line numbers where we have made such claims and we will be happy to make corrections.
>
> Additionally, [14] from OpenAI has shown critiques from a language model, *though imperfect*, is helpful to improve a base language model. Furthermore, [15] from Anthropic has shown that training a harmless AI assistant through *self-improvement leveraging AI feedback is feasible*. Similar from other works we have cited, these are only a few examples where feedback from imperfect LLM has shown to be useful to improve the base model. Our approach aligns with this broader body of work of model self-improvement, emphasizing that while imperfections exist, useful signals can still emerge.
>
> *More broadly, even the entire training pipeline of modern LLMs does not consistently guarantee a “reliable signal.”* The pre-training corpus often contains imperfections, and the accuracy of instruction-following is not assured to be 100%. Similarly, preference data collected from annotators may have inherent biases or inconsistencies. Ongoing research efforts are focused on improving data quality across various stages, including enhancing the reliability and diversity of pre-training and post-training datasets.
>
>
> ---
>
> Additional References:
>
> [14] Saunders, William, et al. "Self-critiquing models for assisting human evaluators." arXiv preprint arXiv:2206.05802 (2022).
>
> [15] Bai, Yuntao, et al. "Constitutional ai: Harmlessness from ai feedback." arXiv preprint arXiv:2212.08073 (2022).

---

> > ### Comment · Reviewer_Gidu · 2024-11-26
> >
> > I would like to thank the reviewers and the authors for this interesting and valuable exchange and I would like to point out that I also fail to understand why the reviewers believe LLMs are used as oracles in this paper. As far as I can tell, the entire evaluation (I.e. success rates) are based on exact evaluation of the agent performance on VWA tasks and not on some unreliable LLM outputs. Using unreliable models to guide a search process is not a novelty nor is it problematic. In fact, it is arguably the only way to solve large complex problems. I would agree with the reviewer that using LLMs to generate the final performance of the agent would be problematic, but this is not what is happening in this work. Even the fact that the verifier (i.e. the value function) is based on LLM outputs is not an issue and quite well aligned with pioneering results in the field that pre-date LLMs: https://www.nature.com/articles/s41586-020-03051-4. The MuZero work is based on a tree-search method that is fully guided by learned, and often unreliable, models of the value function, the policy, and the reward. In the MuZero paper, much as this manuscript, the final results are reported on the exact game evaluations, where they report great improvements even with learned unreliable models.
> >
> > Another important recent work that further supports the validity of this approach is FunSearch, while also dedicating a substantial amount of the manuscript addressing the unreliability of LLMs. https://www.nature.com/articles/s41586-023-06924-6

---

> > > ### Author Response · Authors · 2024-11-27
> > >
> > > We thank reviewer Gidu for engaging in this discussion and appreciate Gidu’s help in providing these additional comments! We would like to confirm that Gidu’s comment on our work is accurate, this includes:
> > >
> > > - “the entire evaluation (I.e. success rates) are based on exact evaluation of the agent performance on VWA tasks… I would agree with the reviewer [XTzE] that using LLMs to generate the final performance of the agent would be problematic, but this is not what is happening in this work”
> > >
> > >    Yes, we only used the ground-truth evaluator in our main experiments to compute the final success rate of different agents, in order to ensure a meaningful comparison between different methods.
> > >
> > > - “Using unreliable models to guide a search process is not a novelty nor is it problematic. In fact, it is arguably the only way to solve large complex problems… Even the fact that the verifier (i.e. the value function) is based on LLM outputs is not an issue and is quite well aligned with pioneering results in the field that pre-date LLMs.”
> > >
> > >    We highly appreciate reviewer Gidu’s comment on this, and we totally agree with this.

---

> > > > ### Comment · Reviewer_XTzE · 2024-11-28
> > > >
> > > > Thanks for the discussion.
> > > >
> > > > MCTS is planning.
> > > > It is based on a model, ideally a perfect model.
> > > > With LLMs, MCTS builds a tree with an imperfect model.
> > > >
> > > > With LLMs, "using multi-agent debate to provide reliable state evaluation", is not based on perfect model.
> > > >
> > > > MuZero is model-based RL.
> > > > MuZero is learning the model, however, with many iterations, following a policy iteration approach.
> > > > It interacts with the environment/ the game, which has the perfect model implicitly.
> > > >
> > > > I do not think we should compare most LLM agent & LLM+MCTS papers with AlphaZero / MuZero:
> > > > Most LLM agent papers are based on a fixed, imperfect LLM, and do not have the improvement loop.
> > > >
> > > > It appears that FunSearch can not guarantee the correctness of the discovered program.
> > > > "Programs generated by the LLM are evaluated and scored on a set of inputs."
> > > > - No LLM can achieve a pass@1 rate of 100%, evaluated with several test cases.
> > > > - Passing several test cases won't guarantee the correctness of code.
> > > >
> > > > Most AI problems are about optimizing some objectives, esp for agent / planning problems.
> > > > Think about the shortest path problem again, AI people won't be satisfied with a feasible / successful solution: any path from the source to the destination.
> > > > The success rate is not a proper objective for AI agents.
> > > >
> > > >
> > > > I adjust the score a little bit, to appreciate the authors' hard work.
> > > > However, I'm concerned with how the LLM community is approaching AI agents.

---

> > > > > ### Author Response · Authors · 2024-12-02
> > > > >
> > > > > > MCTS is planning. It is based on a model, ideally a perfect model… With LLMs [state/reward evaluation] is not based on perfect model
> > > > >
> > > > > We agree that, ideally, MCTS should be done with a perfect world model and a perfect reward model. In the first part of the work, we proposed a variant of *MCTS algorithm* (i.e., R-MCTS), and ran it with a perfect world model (i.e., having access to the web browser) and a non-perfect reward model (i.e., an LLM) to guide the tree search. We note that a ground-truth, perfect reward model that is also capable of providing intermediate rewards is *NOT* available in VisualWebArena (as well as other agent benchmarks such as [16-19]). These benchmarks only offer evaluation scripts that provide a binary outcome at the final state, to evaluate whether the task is successfully completed.
> > > > >
> > > > > In general, we agree and believe that developing a perfect reward model for AI agents (as well as for many other domains such as preference learning) is challenging yet of strong interest to the research community. We leave this for future work.
> > > > >
> > > > >
> > > > > > Think about the shortest path problem again... The success rate is not a proper objective for AI agents.
> > > > >
> > > > > Success rate is a metric proposed by the original VisualWebArena paper [16] to evaluate and compare different agent’s performance. Since solutions for agent tasks *may not be unique*, we believe this is why many agent benchmark authors [17,18,19] used success rate instead of other metrics such as using human demonstrations for exact matching. We followed prior work and used the provided success rate metric for a fair comparison.
> > > > >
> > > > > In general, we agree that success rate may not be the most optimal metric, and suggest that future agent benchmark papers to explore other alternatives.
> > > > >
> > > > >
> > > > >
> > > > > > MuZero is model-based RL. MuZero is learning the model, however, with many iterations... It interacts with the environment/ the game, which has the perfect model implicitly.
> > > > >
> > > > >
> > > > > We are sorry if we misunderstood your statement. We believe during the tree search process, MuZero does *not* interact with an environment but with a learned model. Additionally, the authors also did *not* claim that the their learned model is a prefect one. The abstract from MuZero claims
> > > > >
> > > > > ```
> > > > > … Tree-based planning methods have enjoyed huge success in challenging domains, such as chess and Go, where a perfect simulator is available. However, in real-world problems the dynamics governing the environment are often complex and unknown. In this work we present the MuZero algorithm which, by combining a tree-based search with a learned model, achieves superhuman performance in a range of challenging and visually complex domains, without any knowledge of their underlying dynamics…
> > > > > ```
> > > > >
> > > > > In general, we agree and believe that developing a perfect world model and/or a perfect reward model for AI agents is of strong interest to the research community. We leave this for future work.
> > > > >
> > > > >
> > > > > > Most LLM agent papers are based on a fixed, imperfect LLM, and do not have the improvement loop.
> > > > >
> > > > > In general, we agree that iteratively training the policy/reward model using tree search is beneficial. We note that our work presents a first step towards this direction: we used R-MCTS to improve agent’s performance (Section 4.1), and then *trained GPT-4o using these search data to improve its performance* (Section 4.2). We believe **no prior work** has demonstrated the feasibility of applying AlphaGoZero or MuZero to the domain of AI agents solving computer tasks, and we believe our work serves as a **crucial first step** toward developing approaches alike AlphaGoZero or MuZero.
> > > > >
> > > > >
> > > > >
> > > > >
> > > > > ---
> > > > >
> > > > > References
> > > > >
> > > > > [16] Koh, Jing Yu et al. “VisualWebArena: Evaluating Multimodal Agents on Realistic Visual Web Tasks.” ArXiv abs/2401.13649 (2024): n. Pag.
> > > > >
> > > > > [17] Zhou, Shuyan et al. “WebArena: A Realistic Web Environment for Building Autonomous Agents.” ArXiv abs/2307.13854 (2023): n. pag.
> > > > >
> > > > > [18] Xie, Tianbao et al. “OSWorld: Benchmarking Multimodal Agents for Open-Ended Tasks in Real Computer Environments.” ArXiv abs/2404.07972 (2024): n. pag.
> > > > >
> > > > > [19] Trivedi, Harsh, et al. "Appworld: A controllable world of apps and people for benchmarking interactive coding agents." arXiv preprint arXiv:2407.18901 (2024).

---

> > > > > > ### Comment · Reviewer_XTzE · 2024-12-03
> > > > > >
> > > > > > In AlphaZero / MuZero, MCTS is part of policy iteration, with many iterations.
> > > > > >
> > > > > > AlphaZero and MuZero are for very specific problems, namely games with perfect rules / models.
> > > > > > LLMs are for very general problems, all problems.
> > > > > > This is the fundamental difference.
> > > > > > LLMs + MCTS papers are mimicking AlphaZero / MuZero, but they are very different.
> > > > > >
> > > > > > There are papers applying MCTS to the "inference time scaling", e.g.,
> > > > > > AlphaZero-Like Tree-Search can Guide Large Language Model Decoding and Training, ICML 2024
> > > > > > However, "We conduct initial experiments for one iterative update ...".
> > > > > >
> > > > > > The fact that LLMs people choose success rate as the metric may reflect the limitation of LLMs: LLMs are not trained to optimize some objective of an agentic task.
> > > > > > Usually AI agents look for optimal solutions, not feasible ones.
> > > > > > There are previous works, even many of them, does not justify its validity.

---

> ### Author Response · Authors · 2024-12-03
>
> > In AlphaZero / MuZero, MCTS is part of policy iteration, with many iterations. AlphaZero and MuZero are for very specific problems, namely games with perfect rules / models. LLMs are for very general problems... LLMs + MCTS papers are mimicking AlphaZero / MuZero, but they are very different.
>
> We note that our R-MCTS (Section 3.1) followed by our self-learning (Section 3.2) forms one policy iteration. We agree that works in AI agents is different from works in games with perfect rules, but we believe this does *not* imply LLMs+MCTS is infeasible for AI agents. In fact, our work (and the recent work you cited) shows that MCTS can be applied to LLMs/VLMs to significantly improve their performance (see our Table 1). Augmenting LLMs with MCTS (and our R-MCTS) within an agentic setting is nontrivial, as there are no perfect evaluators/rewards and the environment (i.e., real websites) is highly complex. *We believe it is one of our core contributions and novelty to scale GPT-4o agent’s performance using our R-MCTS algorithm, and then to train and improve GPT-4o’s performance using data obtained during the search process.*
>
>
> We clarify that we did not claim to achieve near-human/super-human performance with our method: we are trying to leverage the generator-discriminator gap to improve our policy model (i.e., GPT-4o) as mentioned in our first response in this discussion. To achieve superhuman performance alike AlphaGo-Zero, we agree and believe iteratively optimizing in an perfect environment with perfect rules/reward is necessary.
>
>
> > The fact that LLMs people choose success rate as the metric may reflect the limitation of LLMs... Usually AI agents look for optimal solutions, not feasible ones. There are previous works, even many of them, does not justify its validity.
>
> We agree that LLMs have limitations. As mentioned in our previous response, we followed the official benchmark implementation to use success rate, and compare against other works with the same metric. We suggest that future agent benchmark papers to explore other alternatives.
>
> However, we believe the work you cited “AlphaZero-Like Tree-Search can Guide Large Language Model Decoding and Training” utilizes final answer accuracy as the metric for GSM8k, which we believe is essentially the same as success rate, since neither ensures “optimality of solutions”.

---

### Official Review · Reviewer_ddLD · 2024-11-04

**Soundness:** 2
**Presentation:** 4
**Contribution:** 3
**Rating:** 6
**Confidence:** 4

**Summary:**

The paper presents a novel algorithm improving the capabilities of VLM agents in long-horizon exploration tasks via a combination of MCTS, and value function estimation by reflection and multi-agent debate. The authors show strong results in the VisualWebArena benchmark as well as showing these behaviors can be distilled back into GPT-4o to improve its base performance.

**Strengths:**

- Well-written and clearly presented paper
- Strong motivation in improving VLM agents in long-horizon tasks
- Novel contrastive reflection proposal that enables a VLM to in-context learn from errors in value estimation
- Thorough ablations over two different value function constructions and individual components of the algorithm
- Informative analysis of the types of errors the agent makes in Section 5.3.

**Weaknesses:**

- Missing relevant multi-step exploration baselines including Reflexion [1] and Intelligent Go-Explore [2]
- The baselines in the main evaluation are not fairly evaluated with equal compute, e.g. does R-MCTS beat ReAct’s 7-shot performance in Table 2? What about ToT’s 2-shot performance? This makes it impossible to evaluate the paper’s contribution. Equally for the self-learning results, would fine-tuning on the best-of-n ReAct trajectories deliver similar improvements in base performance? I would raise my score if the authors could show positive results for this.
- Evaluation is limited to the VisualWebArena benchmark, focusing solely on web navigation tasks. Other VLM domains could include video games, etc.

Minor:
- Line 11: clarify what kind of autonomous agents, the authors clearly mean foundation model based rather than traditional RL
- MCTS should be attributed to Coulom et al. 2006

[1] Reflexion: Language Agents with Verbal Reinforcement Learning Noah Shinn, Federico Cassano, Edward Berman, Ashwin Gopinath, Karthik Narasimhan, Shunyu Yao.

[2] Intelligent Go-Explore: Standing on the Shoulders of Giant Foundation Models Cong Lu, Shengran Hu, Jeff Clune.

**Questions:**

- Why not also fine-tune GPT-4o instead of the in-context contrastive reflection approach?
- Does the proposed algorithm beat the baselines with equal compute?
- What is the impact of the size of the reflection database on value accuracy?

---

> ### Author Response · Authors · 2024-11-17
> **Response 1/2**
>
> Thank you for providing these detailed comments and acknowledging the clarity and quality of our paper’s presentation!
>
> ---
>
> > Missing relevant multi-step exploration baselines including Reflexion [1] and Intelligent Go-Explore [2]
>
> Thank you for the suggestion! We note that Search Agent [1] already outperforms Reflexion (see Table 2 in [1]), and our proposed R-MCTS outperforms Search Agent (see our Table 2). Intelligent Go-Explore (IGE) is based on the Go-Explore algorithm [3], and *offers a different heuristic function* which prioritizes expanding states that are the most “interesting”, instead of states with the highest value (used in ToT and Search Agent) or highest UCT (used in MCTS and R-MCTS). We believe experimenting with different heuristics is a perpendicular direction to our work, which focuses on scaling test-time compute (Section 3.1) and transferring knowledge acquired from search back to the model (Section 3.2).
>
> We will add [1] and [2] in our related work section. We leave the exploration of using different heuristic functions to future work.
>
>
> > The baselines in the main evaluation are not fairly evaluated with equal compute, e.g. does R-MCTS beat ReAct’s 7-shot performance in Table 2?
>
> We clarify that we did not intentionally increase the budget allocated to R-MCTS. All agents (ToT, Search Agent, MCTS, and R-MCTS) are run under the same search budget and hyperparameters such as width and breadth factor (please also see our general response GR-1). However, the greedy best-first nature of algorithms such as ToT BFS/DFS and Search Agent makes them often terminate early as they keep expanding states with high *estimated* value.
>
> Under equal token usage, we show that:
> - Best-of-N ReACT using MAD underperforms R-MCTS under all token usages as shown in the table in GR-2.
> - R-MCTS achieves 35.0% on Classifieds with 4.0x ReACT token usage (from Figure 1 left), whereas Search Agent achieves only 33.8% with 4.2x ReACT token usage (from Table 2).
>
>
> > Equally for the self-learning results, would fine-tuning on the best-of-n ReAct trajectories deliver similar improvements in base performance?
>
> Thank you for the question! We believe you are comparing to the “Best-in-Tree” method, which trains the agent using the best action found during R-MCTS. Since Best-of-7 ReACT underperforms R-MCTS (see table in GR-2), we believe this would result in inferior performance in training.
>
>
> > Evaluation is limited to the VisualWebArena (VWA) benchmark, focusing solely on web navigation tasks. Other VLM domains could include video games, etc.
>
> Thank you for the comment. Many recent work in generalist agents focuses on web benchmarks [1,4,5,6,7] as it offers highly diverse and reproducible environments such as shopping, social forums, collaborative software development, and content management. In addition to the VWA environments we evaluated in this work, we also evaluated GitLab tasks from WebArena, to emphasize on the generality of these browser-based tasks. We present the results below.
>
> | Method          | Search      | Value | GitLab (Tokens) | GitLab (Success) |
> |-----------------|-------------|-------|-----------------|------------------|
> | ReACT           | -           | -     | 1x              | 11.3%            |
> | ToT             | BFS         | SA    | 4.6x            | 14.1%            |
> | ToT             | DFS         | SA    | 4.3x            | 14.1%            |
> | Search Agent    | Best-First  | SA    | 3.0x            | 13.8%            |
> | MCTS            | MCTS        | SA    | 5.7x            | 19.9%            |
> | R-MCTS (ours)   | MCTS        | SA    | 6.0x            | 20.9%            |
> | R-MCTS (ours)   | MCTS        | MAD   | 8.8x            | 23.5%            |

---

> ### Author Response · Authors · 2024-11-17
> **Response 2/2**
>
> > Clarify what kind of autonomous agents on L11, and MCTS should be attributed to Coulom et al. 2006
>
> Thank you for the suggestion! We will clarify on L11 that autonomous agents in this work refers to VLM systems capable of executing computer tasks. We will also cite [8] when mentioning MCTS in this paper.
>
>
> > Why not also fine-tune GPT-4o instead of the in-context contrastive reflection approach?
>
> Sorry if we misunderstood this question. We note that VWA does not provide ground-truth, optimal trajectories for training. We finetuned GPT-4o in Section 4.2 using 1) Best-In-Tree SFT which trains GPT-4o on the actions **returned by a search algorithm (e.g., R-MCTS)**, and Tree-Traversal SFT which trains GPT-4o on all tree traversals that happened during search. Table 3 shows that both training methods recovered over 85% of R-MCTS performance while reducing token usage by up to 4x; and that Tree-Traversal SFT achieves a significantly higher success rate on unseen tasks.
>
>
> > Does the proposed algorithm beat the baselines with equal compute?
>
> Empirically, R-MCTS with the same token budget as Search Agent already out-performs Search Agent. Figure 1 left shows that R-MCTS achieves 35.0% on classifieds with 4.0x ReACT token usage, whereas Search Agent achieves only 33.8% with 4.2x ReACT token usage (from Table 2). Additionally, we also followed your suggestions and compared R-MCTS with Best-of-N ReACT using MAD in the table in GR-2. We find that Best-of-N is much less efficient at scaling test-time compute in agentic tasks compared to R-MCTS with equal compute.
>
>
> > What is the impact of the size of the reflection database on value accuracy?
>
> Thank you for the question. We did not set an upper limit of the reflection database size, but simply controlled the maximum number of reflections to be generated per episode (i.e., 3 per task). In our preliminary experiments, we find varying this number from 1-3 did not significantly impact performance. In general, we believe the size of the database would most strongly influence the performance of the *retrieval model*, which is a perpendicular research direction. We leave this to future work.

---

> ### Author Response · Authors · 2024-11-17
> **References**
>
> References:
>
> [1] Koh, Jing Yu et al. “Tree Search for Language Model Agents.” ArXiv abs/2407.01476 (2024): n. Pag.
>
> [2] Lu, Cong et al. “Intelligent Go-Explore: Standing on the Shoulders of Giant Foundation Models.” ArXiv abs/2405.15143 (2024): n. pag.
>
> [3] Ecoffet, Adrien, et al. "Go-explore: a new approach for hard-exploration problems." arXiv preprint arXiv:1901.10995 (2019).
>
> [4] Wang, Xingyao et al. “OpenHands: An Open Platform for AI Software Developers as Generalist Agents.” (2024).
>
> [5] Zhang, Yao et al. “WebPilot: A Versatile and Autonomous Multi-Agent System for Web Task Execution with Strategic Exploration.” ArXiv abs/2408.15978 (2024): n. Pag.
>
> [6] Xie, Tianbao et al. “OpenAgents: An Open Platform for Language Agents in the Wild.” ArXiv abs/2310.10634 (2023): n. Pag.
>
> [7] Abuelsaad, Tamer et al. “Agent-E: From Autonomous Web Navigation to Foundational Design Principles in Agentic Systems.” ArXiv abs/2407.13032 (2024): n. Pag.
>
> [8] Rémi Coulom. Efficient Selectivity and Backup Operators in Monte-Carlo Tree Search. 5th International Conference on Computer and Games, May 2006, Turin, Italy. ffinria-00116992f

---

> ### Author Response · Authors · 2024-11-19
>
> Thank you again for these questions and feedback! Please let us know at your earliest convenience if you have any further questions or concerns, or would like to us conduct any additional experiments.

---

> > ### Author Response · Authors · 2024-11-20
> >
> > Thank you again for your feedback and thoughtful questions! As we are now halfway through the rebuttal period, I’m following up again to kindly ask if you have any further questions or concerns regarding our responses. If so, please feel free to post them and we will try to answer them as soon as possible!

---

> > > ### Comment · Reviewer_ddLD · 2024-11-23
> > >
> > > Thank you for the response, the new results matching compute and new domain are convincing! I will raise my score.
> > >
> > > Please also match compute for the new domain as well for a fair comparison.

---

> ### Author Response · Authors · 2024-11-24
>
> Thank you for reading through our responses and raising the score! We will incorporate a detailed discussion clarifying our experimental setup and the distinctions between search algorithms in the final manuscript. For completeness, we also address the additional question below.
>
> ---
>
> > Please also match compute for the new domain as well for a fair comparison.
>
> Since it is difficult to precisely control different search algorithms token usage (as they can produce trees of different shapes, see our GR-1 for more details), we followed your other suggestions and **matched our R-MCTS_{MAD} with Best-of-N ReACT**. We present the updated table below (changes shown in *italic*). Similar to our conclusion in GR-2, we find Best-of-N underperforms R-MCTS.
>
> | Method          | Search      | Value | GitLab (Tokens) | GitLab (Success) |
> |-----------------|-------------|-------|-----------------|------------------|
> | ReACT           | -           | -     | 1x              | 11.3%            |
> | ToT             | BFS         | SA    | 4.6x            | 14.1%            |
> | ToT             | DFS         | SA    | 4.3x            | 14.1%            |
> | Search Agent    | Best-First  | SA    | 3.0x            | 13.8%            |
> | MCTS            | MCTS        | SA    | 5.7x            | 19.9%            |
> | *Best-of-9 ReACT*            | -        | *MAD*    | *9.0x*            | *15.3%*            |
> | R-MCTS (ours)   | MCTS        | SA    | 6.0x            | 20.9%            |
> | R-MCTS (ours)   | MCTS        | MAD   | 8.8x            | 23.5%            |

---

> > ### Comment · Reviewer_ddLD · 2024-12-03
> >
> > Great, good luck with your submission!

---

### Official Review · Reviewer_Gidu · 2024-11-11

**Soundness:** 3
**Presentation:** 3
**Contribution:** 2
**Rating:** 6
**Confidence:** 3

**Summary:**

This paper provides an overview of MCTS based methods for improved test-time (i.e. inference-time) algorithms for Visual Language Models and tested on the VisualWebArena benchmark for complex web environments. This benchmark is particularly important as it poses a substantial challenge, as VLMs have been shown to often fall short of human-level performance in such tasks, hence any progress in this space is important. The most performant variation of the search algorithm studied in this paper is R-MCTS, which leverages the well known benefits of the traditional Monte Carlo Tree Search (MCTS) algorithm, popular method for improving decision-making in long-horizon tasks with a large action space. Although MCTS has been used in other contacts with a learned world model, this paper is using the WebArena simulator to expand tree nodes. None of the results use the VLM or some other learned model for transitions. VLM is only used for policy and value functions.

R-MCTS extends traditional MCTS by incorporating contrastive reflection and utilizing multi-agent debate. Contrastive reflection allows agents to learn from past interactions and dynamically improve their search efficiency. Perhaps the most impactful design choice is the Memorization of reflection. Reflection embeddings are stored in a vector database and, during inference, relevant reflections are retrieved from the database and appended to the agent's current context, enabling it to learn from past mistakes and improve future task executions. On the other hand, Multi-Agent Debate prompts multiple VLMs to generate individual value estimates, which are then aggregated to produce a final estimate. This approach aims to provide a more holistic view of the current state and encourage stronger reasoning through collaborative/adversarial incentives.

Experiments performed on the VisualWebArena (VWA) benchmark, which evaluates multimodal agents' performance across various web navigation tasks, demonstrated the effectiveness of R-MCTS, with significant performance boosts over alternative search methods.
Moreover, the authors demonstrate that the search trajectories can be effectively transferred back to base models via fine-tuning, resulting in improved performance without requiring search augmentation at test time.

**Strengths:**

Building agents that are capable of solving complex web navigation using methodologies that are as general as possible will prove to be among the most impactful contributions in the field of AI. The authors are doing a good job to emphasize this and they provide an important contribution in this area, properly aligned with many findings that emphasize the strong VLM capabilities in inference time reasoning and planning.Additionally, the authors demonstrated how easily integrated concepts drawn from human dialogue can be used to improve generic methods. They did this by pinpointing two intuitive modifications to MCTS: contrastive reflection and debating.The boost in performance with the proposed methods is also quite significant, raising the bar on the type of capabilities that can be achieved with the use of LLMs, while also demonstrating the strong potential of scaling up these methods (e.g. longer search horizons, more inference time computations). Lastly, I will point out that the paper is overall well contextualized with respect to the current literature on agentic LLM frameworks, search-based methods for LLMs/VLMs and other inference time algorithms. I would also like to applaud the contributions to SFT-based experiments and emphasizing the importance of generating quality synthetic data, given the cost and logistical challenges associated with human data collection

**Weaknesses:**

I will perhaps start with what I believe is one of the most important limitations of the work: the use of agent-environment interactions to generate and expand new nodes in the search tree. Unless there is something that I am missing, the search algorithm is based on Alpha-Zero and not Mu-Zero, which essentially means that the results rely heavily on access to a simulator at test time. This should be stated explicitly and the experimental work should be adjusted to account for this rather important detail. In particular, neither ReAct,  nor TreeOfThought (albeit it could be adapted to use the simulator) are presented as a way to enhance agent performance by using an exact simulator. It should be very clear for each of the methods in the experimental section whether the method uses the simulator and if so, to what extent (e.g. how many transitions). Note that I am a strong proponent of using a simulator when available and I also think this should be clarified as a distinct paradigm from a method similar to MuZero, which is fully devoted to a simulator, and hence much more generalizable. At the same time, having access to ground truth transitions can indeed make a huge difference in terms of access to important information, hence empirical results should be presented in a way that makes this use much more transparent. Note that many results in Reinforcement Learning, where agents similarly have access to a simulator, will always report results as functions of “episodes” or “epochs”, both of which are essentially reporting the amount of simulated interactions that an agent has access to. As such, comparing an agent such as ReAct that (AFAICT) doesn’t have access to these simulated transitions, or even another search agent for which readers have no clear understanding of how it works, can be quite misleading.

Figure 1 (left) should be explained much more clearly and that should be done in the captions of the Figure, given that the figure is placed in on the first page. It is great that the reader is given the impact of the work as early as possible. I personally struggled to understand the result both when I first saw it on the first page and after I reached the experimental section and the Figure was explained. First, if the number of tokens for ReAct is not something that can be controlled, then the plot in Figure 1 (a) needs another baseline, e.g. MCTS, Search-Agent. As it stands, the result is stating that the more an agent has access to a simulator to learn about possible trajectories, the better the performance - but I imagine the authors want a stronger message. Similarly, for the Figure on the right side, the plot now discusses actions allowed and “unseen” success rate, which once again seems quite selective and there is no clear message takeaway message. Do you want to convey the fact that SFT on search trajectories allows the model to learn the shortest path?

I would also like to raise concerns about the improvement step, which is essentially doubling down on the advantage of using a simulator by storing embeddings of reflections. This is akin to Replay buffers in Reinforcement Learning. Once again, this seems to be essential to your method, but there is no reason why this improvement step cannot be applied for the other search methods.

Given my take on the main results (i.e. Figure 1) and my comments on the simulator, I would encourage the authors to be more explicit about the significance of the work. As it stands, the message is mostly centered around R-MCTS_{MAD} improving SoTA on VWA, but focus should be given to agents that can utilize the simulator. Obvious candidates for these agents include algorithms like BFS, DFS, and MCTS, and the additional components the authors propose should then be clearly justified and ablated, especially when they can be applied independent of the search method. Note that the same I argument can be made for the multi-agent value function.

For experiments, perhaps testing more than two underlying models is necessary to better understand the phenomenon of search for VWA, albeit it is understandable if this incurs costs that are too high for their corresponding institution/s. Additionally, empirical results should be much more explicit on the number of simulations that were used in each case. Even for results on SFT where the number of simulations is perhaps irrelevant at test time, it should be clear how many simulations were necessary to build the datasets used for SFT.

I noticed that there is a very clear emphasis on the number of tokens used by either of the methods. I wonder if the authors considered Best-on-N (i.e. generate many candidate trajectories without any search) as a baseline, and if so, why was it not included in the studies? To my point on scaling results such as those presented in Figure 1a, it is important to acknowledge that, as we increase the number of available tokens, there are things simpler than MCTS that could be done and search should only be justified when the performance gap is significant. Additionally, why can’t we increase the size of the search tree for ToT, Search-Agents, MCTS, to match the token consumption of R-MCTS?

Lastly, as far as novelty is concerned, I stand by my take that the paper is a very interesting study of search methods when a simulator for VWA is provided to the agent and the R-MCTS algorithm is a simple twist to the very popular MCTS. The modifications themselves, whether contrastive reflections, multi-agent debates, or self learning, are, as the authors already made clear, existing concepts with straightforward incorporation with MCTS on VLMs.

**Questions:**

Since my stance on the paper is strongly grounded on my understanding that the trees for the search algorithms are built with full access to the VMA simulators, I would love if the authors are more descriptive on how experiments were conducted. For context, my assumption is based on the description of the MCTS algorithm in the Appendix, where $\mathcal{T}$ is used to expand new nodes.

---

> ### Author Response · Authors · 2024-11-17
> **Response 1/2**
>
> Thank you for providing these detailed feedbacks and acknowledging our “interesting study of search methods”!
>
> ---
>
> > One of the most important limitations of the work: the use of agent-environment interactions to generate and expand new nodes in the search tree… It should be very clear for each of the methods in the experimental section whether the method uses the simulator and if so, to what extent.
>
> We clarify that we followed prior work [1] where **all search methods** (including ToT BFS, ToT DFS, Search Agent, MCTS, and R-MCTS) have **full** access to a web browser instance but **no** access to the ground-truth evaluator/reward function. We believe this is a practical setup since it does not require any human intervention/related resources, other than accessing a web browser.
>
> We are sorry for the confusion, and will clarify this setup more in our Section 4.
>
>
> > [Figure 1 right] discusses actions allowed and “unseen” success rate … Do you want to convey the fact that SFT on search trajectories (i.e., Tree-Traversal SFT) allows the model to learn the shortest path?
>
> Sorry for the confusion. We did not intend to convey that Tree-Traversal SFT (TrT SFT) trains the model to learn the shortest path. TrT SFT trains the agent to learn to **explore and backtrack**, a feature of the MCTS-based algorithms. Our result in Figure 1 right shows that agents after TrT SFT improves performance when more action steps are allocated, similar to scaling token usages with R-MCTS in Figure 1 left. This indicates that TrT SFT is **not** teaching the agent the shortest path (otherwise performance would change with an increased action budget), but rather how to conduct test-time search. Figure 4 provides a concrete example of TrT SFT trained GPT-4o exploring, evaluating, and backtracking without augmenting with search algorithms.
>
>
> > Concerns about the improvement step… Once again, this seems to be essential to your method, but there is no reason why this improvement step cannot be applied to the other search methods.
>
> Thank you for pointing this out! We did not state that our training methods (e.g., Tree-Traversal SFT) are specific to our R-MCTS algorithm. In fact, our methods can be applied to any search algorithm that iteratively constructs a tree. In this work, we chose R-MCTS as an example (see L240) simply because it has the best performance, which is essential for an effective self-learning setup (Section 3.2).
>
> We will add this discussion in our Section 3.2.
>
>
> > As it stands, the message is mostly centered around R-MCTS_{MAD} improving SoTA on VWA, but focus should be given to agents that can utilize the simulator. Obvious candidates for these agents include algorithms like BFS, DFS, and MCTS.
>
> We are sorry if we misunderstood this question. We clarify that in our experiments (Section 4.1) all search methods (ToT BFS, ToT DFS, Search Agent, MCTS, and R-MCTS) have full access to a browser instance to perform simulation. **No** methods have access to ground-truth reward function or any human-related signals. In addition, we also use the same hyperparameters (e.g., breadth and width factor, see L306-310) different search algorithms to ensure a fair comparison.
>
> Please see our GR-1 and GR-2 for more details.
>
>
> > Additionally, empirical results should be much more explicit on the number of simulations that were used in each case. Even for results on SFT where the number of simulations is perhaps irrelevant at test time, it should be clear how many simulations were necessary to build the datasets used for SFT.
>
> We are sorry if we misunderstood this question. We reiterate that all search methods only have access to a web browser and have **no** access to the ground-truth reward function or any human-related signals. The number of simulations (e.g., nodes) is proportional to the tokens reported in our experiment. The SFT training data has 10.34 nodes per task, and they *directly come from the R-MCTS search trees from Table 2*. In general, we believe it is difficult to estimate *a priori* the minimum number of simulation/nodes to build an effective SFT dataset. We leave this for future work.

---

> ### Author Response · Authors · 2024-11-17
> **Response 2/2**
>
> > I wonder if the authors considered Best-on-N (i.e. generate many candidate trajectories without any search) as a baseline… To my point on scaling results such as those presented in Figure 1a, it is important to acknowledge that, as we increase the number of available tokens, there are things simpler than MCTS
>
> Thank you for pointing this out! Please refer to our GR-2, where we showed that Best-of-N ReACT is much less efficient at scaling test time compute than search methods such as R-MCTS. We will add this result in our final manuscript.
>
>
> > If the number of tokens for ReAct is not something that can be controlled, then the plot in Figure 1 (a) needs another baseline
>
> Yes, controlling token usage of different search algorithms is difficult. We implemented your suggestion above, and compared Best-of-N to R-MCTS across different token costs. Since ground-truth evaluator cannot be accessed during search time, Best-of-N becomes an inefficient version of search algorithms such as R-MCTS, as it rollouts the trajectory *regardless of the quality of the states*. Please refer to our Table in GR-2 for this result.
>
> We will follow your suggestion and update Figure 1 (a) with this Best-of-N ReACT results.
>
>
> > The result [in Figure 1 left] is stating that the more an agent has access to a simulator to learn about possible trajectories, the better the performance - but I imagine the authors want a stronger message.
>
> The first part of the work focuses on exploring how to scale test-time compute to improve VLM agents performance (Figure 1 left). Naive methods to scale up compute (e.g., Best-of-N ReACT) cannot efficiently improve performance compared to search algorithms such as R-MCTS, which we show in the table in GR-2. Similarly, in Table 2 in this paper we also show that other search algorithms from prior work (ToT BFS, ToT DFS, and Search Agent) also underperforms R-MCTS.
>
>
> > Why can’t we increase the size of the search tree for ToT, Search-Agents, MCTS, to match the token consumption of R-MCTS?
>
> We clarify that all search methods (ToT BFS, ToT DFS, Search Agent, MCTS, and RMCTS) are ran under the same search budget, and the token variations are caused by the agent themselves terminate quickly with incorrect states, as they focus solely on rolling out states *perceived* to have high value (see L445-448). Please refer to GR-1 for more details.
>
> **Empirically, R-MCTS with the same token budget as Search Agent already out-performs Search Agent**. Figure 1 left shows that R-MCTS achieves 35.0% on classifieds with 4.0x ReACT token usage, whereas Search Agent achieves only 33.8% with 4.2x ReACT token usage (from Table 2). Additionally, naively scaling up compute with Best-of-N ReACT also underperforms R-MCTS under the same token consumption, which we show in GR-2.

---

> ### Author Response · Authors · 2024-11-17
> **References**
>
> References:
>
> [1] Koh, Jing Yu et al. “Tree Search for Language Model Agents.” ArXiv abs/2407.01476 (2024): n. Pag.

---

> ### Author Response · Authors · 2024-11-19
>
> Thank you again for these questions and feedback! Please let us know at your earliest convenience if you have any further questions or concerns, or would like to us conduct any additional experiments.

---

> > ### Comment · Reviewer_Gidu · 2024-11-19
> >
> > Thank you for the very useful clarifications and for running the additional experiments that were suggested. These are quite valuable and indeed are moving much closer to a proper ablation study. As per our exchange, it is quite clear that the manuscript needs to include more information on the multiple sources of improvement (e.g. more tokens, more simulations, etc), as well as clear limitations in some instances (e.g. hard to control the number of tokens for some of the methods). Having a proper update to Figure 1a, where different methods (including Best-of-N for ReAct, or Best-of-N for ToT) are mapped alongside R-MCTS_{MAD} will give a much more clear idea of the benefits of the method provided in the manuscript.
> >
> > Perhaps one aspect that still seems to be missing for a fair comparison Best-of-N with MAD as evaluation - please clarify whether you used MAD vs. SA to pick the best of the candidates generated by ReAct. Since the value function is interchangeable with these methods, it is important to do a proper side-by-side on where the contributions come from.
> >
> > For the Figure 1b, please update the discussion for added clarity on the benefits of SFT. If the number of actions allowed is related to the number of tokens, please change the x-axis to be consistent to Figure 1a and consistent with your message (i.e. SFT makes search more efficient).
> >
> > To your point on that "it is difficult to estimate a priori the minimum number of simulation/nodes to build an effective SFT dataset. We leave this for future work." - I agree and I think you can still report the number that have been used in your experiments for a clear picture of how important  to a simulator for the reported results.
> >
> > Perhaps I will also further clarify my comments on "having access to a simulator". First, it is great to have clarity on the fact that all methods have access to one, I am also happy to know that the authors will add more numerical clarity on how much this is being used by every one of the methods, and indeed it is important that we observe these results for instances where there is no access to human feedback. Second, I will re-iterate my comment that these results are still in the realm of Alpha-Zero like search, as opposed to Mu-Zero. Over relying on simulators can lead to methods that will not generalize outside the scope of these limited benchmarks and the community needs to have a clear handle on how much of the LLM-inference cost is trade-off for access to simulators (which is what many search methods do). Since the authors are committed to include more results and explanations that help readers navigate this nuance, I will increase my score to a 6.

---

> > > ### Author Response · Authors · 2024-11-20
> > > **Response to further questions/concerns**
> > >
> > > Thank you for reading through our responses, and thank you for raising your score! Based on your feedback, we identified a few key points that perhaps need further clarification. We address them below.
> > >
> > > ---
> > >
> > > > The manuscript needs to include more information on the multiple sources of improvement (e.g. more tokens, more simulations, etc)
> > >
> > > Thank you for the comment! We believe you are referring to the different token usage by search methods (which is proportional to the number of simulations), as we believe there are no other differences.
> > >
> > > Here we provide a few clarifications in case you were referring to any of the following:
> > > - All search methods used the exact same prompts for expansion and evaluation (except for R-MCTS_MAD which uses MAD as value function), which is described in L293-304.
> > > - All search methods used the same value function logic, except for the last row in Table 2 where we used our new MAD for R-MCTS to further improve performance. We note that R-MCTS + SA also outperforms all other methods.
> > > - Additional components introduced in R-MCTS (contrastive self-reflection and the proposed MAD value function) are "outside" of the search logic in MCTS, which we carefully ablated in Section 5.2.
> > >
> > > Finally, we clarify that **despite drastically different names** (ToT, Search Agent, MCTS), the main algorithmic difference is only **selecting which node to expand/explore next**. ToT and Search Agent use a greedy, best-first strategy, while MCTS and R-MCTS uses UCT to balance exploration and exploitation. This difference is **NOT** part of our contributions in the R-MCTS algorithm.
> > >
> > > Thank you for mentioning this. We will clarify this in our baseline section (L285-298).
> > >
> > >
> > > > Clear limitations in some instances (e.g. hard to control the number of tokens for some of the methods)
> > >
> > > Thank you for the comment! We clarify that the search methods we compared against (ToT BFS, ToT DFS, Search Agent, MCTS) are from prior work (see L285-291) and their algorithmic behavior is beyond our control. For a fair comparison, we did set the same search budget and hyperparameters in our main experiments (see GR-1). We emphasize that precisely controlling the token usage *a priori* is infeasible for all search algorithms, as it is hard to estimate which action the algorithm will generate and hence explore beforehand.
> > >
> > >
> > > > Having a proper update to Figure 1a, where different methods (including Best-of-N for ReAct, or Best-of-N for ToT) are mapped alongside R-MCTS_{MAD} will give a much more clear idea of the benefits of the method provided in the manuscript.
> > >
> > > Thank you for the suggestion. Since the table in GR-2 already provides the full result of Best-of-N ReACT, we will add it in our final manuscript.
> > >
> > > Since Best-of-N can be seen as a search algorithm itself [4], we believe Best-of-N + ToT constitutes an “ensemble” of search algorithms. This is a different but very interesting idea (e.g., one can also ensemble Best-of-N + Search Agent, Best-of-N + MCTS, etc.), which we would like to leave for future work to explore.
> > >
> > > > One aspect that still seems to be missing for a fair comparison of Best-of-N with MAD as evaluation - please clarify whether you used MAD vs. SA to pick the best of the candidates generated by ReAct.
> > >
> > > We are sorry if we misunderstood your question. In the table in GR-2, *we indicated that MAD is used (see the “value” column)* to pick the best ReACT trajectory. To prevent misunderstanding, we note that non-search methods such as ReACT do not use value functions within a trajectory (since they only go forward). Best-of-N ReACT only uses the value function at the very end to pick the best trajectory, whereas search methods such as R-MCTS also uses it during search.
> > >
> > > Please feel free to pose further questions in case we misunderstood this.
> > >
> > >
> > > > Perhaps I will also further clarify my comments on "having access to a simulator"... I will reiterate my comment that these results are still in the realm of Alpha-Zero like search, as opposed to Mu-Zero.
> > >
> > > We agree that implementing a MuZero-like algorithm is highly exciting, where an additional VLM could model website transitions (e.g., predicting the next webpage after a search query). In this work, we followed many recent agent-related work [1,2,3] and allowed search methods to use an actual browser. We believe this is primarily due to the poor performance of current VLM models in these computer tasks [5,6], and that developing simulator-based search methods like AlphaGo Zero is a *crucial first step* toward building MuZero-like approaches.
> > >
> > > We want to thank the reviewer again for this suggestion. We emphasize that it is also our research goal to build more general and performant agents like Mu-Zero. We will study this closely in our future work.

---

> > > > ### Author Response · Authors · 2024-11-20
> > > > **References for "Response to further questions/concerns"**
> > > >
> > > > [1] Koh, Jing Yu et al. “Tree Search for Language Model Agents.” ArXiv abs/2407.01476 (2024): n. Pag.
> > > >
> > > > [2] Zhou, Andy et al. “Language Agent Tree Search Unifies Reasoning Acting and Planning in Language Models.” ArXiv abs/2310.04406 (2023): n. Pag.
> > > >
> > > > [3] Zhang, Yao et al. “WebPilot: A Versatile and Autonomous Multi-Agent System for Web Task Execution with Strategic Exploration.” ArXiv abs/2408.15978 (2024): n. pag.
> > > >
> > > > [4] Snell, Charlie et al. “Scaling LLM Test-Time Compute Optimally can be More Effective than Scaling Model Parameters.” ArXiv abs/2408.03314 (2024): n. Pag.
> > > >
> > > > [5] Xie, Tianbao et al. “OSWorld: Benchmarking Multimodal Agents for Open-Ended Tasks in Real Computer Environments.” ArXiv abs/2404.07972 (2024): n. Pag.
> > > >
> > > > [6] Koh, Jing Yu et al. “VisualWebArena: Evaluating Multimodal Agents on Realistic Visual Web Tasks.” ArXiv abs/2401.13649 (2024): n. pag.

---

### Author Response · Authors · 2024-11-17
**General Response**

We thank all reviewers for their helpful feedback and suggestions!

---

> GR-1: Concerns about experimental setup: the use of agent-environment interactions/more tokens/etc by RMCTS agents

We clarify that in our Table 2, all search methods (ToT BFS, ToT DFS, Search Agent, MCTS, and RMCTS) are ran under the same search budget, have full access to a browser instance to perform simulation, and uses a self-evaluated value function. **No** methods have access to ground-truth reward function or any human-related signals. In addition, we also use the same hyperparameters (e.g., breadth and width factor, see L306-310) different search algorithms to ensure a fair comparison. We believe this corresponds to the traditional model self-improvement setup where the agent has **no access** to any human-related signals but only data created by itself.

The variation in token usage arises from agents terminating the search either when they **believe they have achieved the task** (with an estimated value of 1.0) or when the **search budget is exhausted**, whichever comes first. In practice, we find best-first methods such as ToT and Search Agent often terminate quickly with incorrect states, as they focus solely on rolling out states *perceived* to have high value (see L445-448).

We will add these clarifications in our experimental setup section (4) and our analysis section (5.3).


> GR-2: Have the authors best-on-N (e.g., with ReACT) as a much simpler method to scale token consumption for performance?

Best-of-N with ReACT rolls out N trajectories, and returns the best trajectory scored by a reward model/evaluator. However, since no methods can access the ground-truth reward (see response above), best-of-N ReACT with an estimated reward function (e.g., MAD) becomes an inefficient version of the tree search used in this work. This is because best-of-N rolls out the trajectory *regardless of state quality*, whereas MCTS (and R-MCTS) prioritizes visiting states using an exploration-exploitation trade-off (the UCT bound, see L121-124).

Empirically, we show this by running best-of-N ReACT with various N on Classifieds, and compare it against R-MCTS:


| Method            | Value | Token | Success |
|--------------------|-------|-------|---------|
| ReACT             | -     | 1x    | 28.6%   |
| Best-of-4 ReACT   | MAD   | 4x    | 32.5%   |
| R-MCTS            | MAD   | 4x    | **35.0%** |
| Best-of-7 ReACT   | MAD   | 7x    | 35.0%   |
| R-MCTS            | MAD   | 7x    | **37.6%** |
| Best-of-10 ReACT  | MAD   | 10x   | 37.6%   |
| R-MCTS            | MAD   | 10x   | **43.6%** |
| Best-of-12 ReACT  | MAD   | 12x   | 38.9%   |
| R-MCTS            | MAD   | 12x   | **47.5%** |

Results above show that R-MCTS is much more efficient at scaling test-time compute at agentic tasks, especially when more tokens are allocated.

---

### Author Response · Authors · 2024-12-03

**We thank all reviewers for their valuable comments and their engagement throughout the discussion period.** We are glad that reviewers found the paper to be well presented and contextualized (Gidu, ddLD, XTzE), the approach to be novel (ddLD, SSY4) and impactful (Gidu), and experiments to be thorough (ddLD) and extensively evaluated (SSY4).

Throughout the discussion period, we believe that our responses addressed the concerns brought up in each review, and we have updated our manuscript to reflect these changes (shown in red). Please let us know if there is anything else we can further clarify before the end of the discussion phase.

---

We summarize the main concerns and how we addressed them below.

\
> Search budget and token usage variation of different search algorithms

We clarified that all search methods (ToT BFS, ToT DFS, Search Agent, MCTS, and RMCTS) are run under the same search budget and hyperparameters, have full access to a browser instance to perform simulation, and a self-evaluated value function. We emphasize that precisely controlling the token usage apriori is infeasible for all search algorithms, as it is hard to estimate beforehand which action the algorithm will generate/explore and whether the agents will terminate early once they believe they have achieved the task. To this end, we followed reviewer Gidu and ddLD’s suggestions and additionally ran Best-of-N ReACT, and showed that under equal token R-MCTS substantially outperforms Best-of-N ReACT (response GR-2).

We believe we have addressed the concern throughout the discussion, and we have updated Figure 1 and Section 4-4.3 in our manuscript correspondingly.

\
> Access to a simulator or the ground truth evaluator code

We clarified that all search methods have full access to a web browser instance but **no access** to the ground-truth evaluator/reward function. We believe this is a practical setup since it does not require any human intervention/related resources, other than accessing a web browser. We agree with reviewer Gidu that implementing a MuZero-like algorithm (using an additional VLM to model website transitions) is highly exciting. In this work, we followed many recent agent-related work [1,2,3] and allowed search methods to use an actual browser. We believe that developing simulator-based search methods like AlphaGo Zero is a *crucial first step* toward building MuZero-like approaches. We emphasize that it is also our research goal to build more general and performant agents like Mu-Zero. We will study this closely in our future work.

We believe we have addressed the concern, and we have added these clarifications in Section 4 of our manuscript. We also added a discussion on the idea of developing MuZero-like approaches in our limitations section.

\
> Whether our work required access to the model of the environment/can generalize to other domains

We clarified that all agents only access a visual or textual representation of the environment state (e.g., a screenshot) as input, and *is independent of the backend web implementation*. All methods do **not** have access to any unrealistic web resources. This makes it easy to transfer between many tasks and domains, which we demonstrated by additionally experimenting on the GitLab domain (see response to reviewer ddLD and SSY4).

We believe we have addressed the concern, and we have added clarifications in Section 2.3. We have also updated our Table 1 with these GitLab results.

\
> Model self-improvement using self-evaluations/reward from an VLM is not a valid approach, although the algorithm may improve the performance.

We agree that, ideally, search and training should be conducted with perfect evaluator/reward models. However, this is hard to obtain in many domains, and many current research [4,5,6,7,8] relies on imperfect evaluation/reward models to further advance state-of-the-art. Reviewer Gidu shared a similar stance: “Using unreliable models to guide a search process is not a novelty nor is it problematic. In fact, it is arguably the only way to solve large complex problems.”

We understand reviewer XTzE’s concern towards the whole community: “I'm concerned with how the LLM community is approaching AI agents”. In general, we agree and believe that developing a reliable and near-perfect reward model for AI agents (as well as for many other domains such as preference learning) is challenging yet of strong interest to the research community. We leave this for future work.

\
> Other individual concerns/questions

We believe we have also addressed other individual questions, such as whether our Tree-Traversal SFT is designed specifically for R-MCTS, and adding standard deviations to our Table 1. We have updated our manuscript to reflect these questions and discussions.

---

> ### Author Response · Authors · 2024-12-03
> **References**
>
> [1] Koh, Jing Yu et al. “Tree Search for Language Model Agents.” ArXiv abs/2407.01476 (2024): n. Pag.
>
> [2] Zhou, Andy et al. “Language Agent Tree Search Unifies Reasoning Acting and Planning in Language Models.” ArXiv abs/2310.04406 (2023): n. Pag.
>
> [3] Zhang, Yao et al. “WebPilot: A Versatile and Autonomous Multi-Agent System for Web Task Execution with Strategic Exploration.” ArXiv abs/2408.15978 (2024): n. pag.
>
> [4] Gulcehre, Caglar et al. “Reinforced Self-Training (ReST) for Language Modeling.” ArXiv abs/2308.08998 (2023): n. pag.
>
> [5] Jiaxin Huang, Shixiang Gu, Le Hou, Yuexin Wu, Xuezhi Wang, Hongkun Yu, and Jiawei Han. 2023. Large Language Models Can Self-Improve. In Proceedings of the 2023 Conference on Empirical Methods in Natural Language Processing, pages 1051–1068, Singapore. Association for Computational Linguistics.
>
> [6] Aksitov, Renat et al. “ReST meets ReAct: Self-Improvement for Multi-Step Reasoning LLM Agent.” ArXiv abs/2312.10003 (2023): n. Pag.
>
> [7] Sun, Zhiqing et al. “SALMON: Self-Alignment with Instructable Reward Models.” International Conference on Learning Representations (2023).
>
> [8] Hu, Chi et al. “Teaching Language Models to Self-Improve by Learning from Language Feedback.” Annual Meeting of the Association for Computational Linguistics (2024).

---

### Meta-Review · Area_Chair_2GAu · 2024-12-19

**Metareview:**

This paper proposes an MCTS approach that can update a base foundation model on long horizon tasks. This is clearly highly relevant right now and the authors did a good job of improving the evaluations during the rebuttal phase. I think the work will be interesting for many communities given the current emphasis on agents, and recommend acceptance.

**Additional Comments On Reviewer Discussion:**

Reviewer XTzE did not engage in an appropriate manner, stating questionable claims as fact to discount the work. The other reviewers had a healthy discussion, for example Reviewer ddLD raised their score after an improved experimental protocol was added to the paper.

---

### Decision · Program_Chairs · 2025-01-22

Accept (Poster)